# 🔖 xFinder: Large Language Models as Automated Evaluators for Reliable Evaluation

**Qingchen Yu**[1][*]   **Zifan Zheng**[1][*]   **Shichao Song**[2][*]   **Zhiyu Li**[1][†]
**Feiyu Xiong**[1]   **Bo Tang**[1]   **Ding Chen**[1]
[1]Institute for Advanced Algorithms Research, Shanghai
[2]Renmin University of China
`lizy@iaar.ac.cn`

## Abstract

The continuous advancement of large language models (LLMs) has brought increasing attention to the critical issue of developing fair and reliable methods for evaluating their performance. Particularly, the emergence of cheating phenomena, such as test set leakage and prompt format overfitting, poses significant challenges to the reliable evaluation of LLMs. As evaluation frameworks commonly use Regular Expression (RegEx) for answer extraction, models may adjust their responses to fit formats easily handled by RegEx. Nevertheless, the key answer extraction module based on RegEx frequently suffers from extraction errors. Furthermore, recent studies proposing fine-tuned LLMs as judge models for automated evaluation face challenges in terms of generalization ability and fairness. This paper comprehensively analyzes the entire LLM evaluation chain and demonstrates that optimizing the key answer extraction module improves extraction accuracy and enhances evaluation reliability. Our findings suggest that improving the key answer extraction module can lead to higher judgment accuracy and improved evaluation efficiency compared to the judge models. To address these issues, we propose xFinder, a novel evaluator for answer extraction and matching in LLM evaluation. As part of this process, we create a specialized dataset, the **K**ey **A**nswer **F**inder (KAF) dataset, to ensure effective model training and evaluation. Generalization tests and real-world evaluations show that the smallest xFinder model, with only 500 million parameters, achieves an average extraction accuracy of 93.42%. In contrast, RegEx accuracy in the best evaluation framework is 74.38%. The final judgment accuracy of xFinder reaches 97.61%, outperforming existing evaluation frameworks and judge models. All resources for xFinder are available at `https://github.com/IAAR-Shanghai/xFinder`.

## 1 Introduction

Large language models (LLMs) have seen rapid advancements in recent years (Brown et al., 2020; Achiam et al., 2023). To explore the strengths, weaknesses, and differences of various LLMs, numerous researchers have introduced a variety of evaluation benchmarks aimed at evaluating the performance of LLMs across various dimensions (Talmor et al., 2019; Clark et al., 2018; Hendrycks et al., 2020; Cobbe et al., 2021). Furthermore, to facilitate comparisons of different LLMs on these benchmarks, several institutions have developed unified LLM evaluation frameworks, such as LM Eval Harness[1] (Gao et al., 2021) and OpenCompass (OpenMMLab, 2023). These frameworks are capable of uniformly evaluating LLM in reasoning, text generation, and domain-specific knowledge.

Despite the widespread application of the aforementioned evaluation benchmarks and frameworks, some researchers have raised concerns about the fairness and reliability of current LLM evaluation efforts (Aiyappa et al., 2023; Ye et al., 2024; Yu et al., 2024). For instance, the benchmarks may suffer from data contamination and leakage in their test sets (Zhou et al., 2023; Oren et al., 2023).

---

[*]Equal contribution. † Corresponding authors.
[1]LM Eval Harness is the backend for Hugging Face's Open LLM Leaderboard (Beeching et al., 2023).

Furthermore, a systematic review has revealed multiple unreliabilities at various stages of the existing evaluation pipeline, as illustrated in Figure 1.

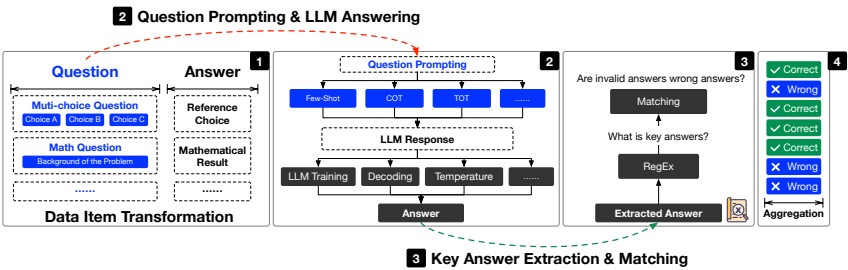

Figure 1: Typical LLM Evaluation Pipeline.

**Data Item Transformation** Given the uncertainty of LLM output content, evaluation frameworks often transform datasets to achieve uniform evaluation across different tasks. For example, they may convert datasets into the Multiple-Choice Question-Answer (MCQA) with alphabet options. However, it has been shown that for some tasks, the evaluation results under this MCQA form with alphabet options may differ from those under the open-ended setting (Röttger et al., 2024).

**Question Prompting and LLM Answering** Various institutions may employ different configurations (such as temperature settings, decoding strategies, etc.) and prompts when evaluating LLMs. LLMs are highly sensitive to these settings (Zhou et al., 2023; Zheng et al., 2025). Even minor modifications (such as changing the case of letters in options or adding extra spaces) can significantly alter the model's responses (Röttger et al., 2024; Gu et al., 2023; Sun et al., 2023; Yukun et al., 2024; Chen et al., 2024), thus affecting the fairness of the evaluation outcomes. Moreover, while employing consistent prompts may ensure a degree of fairness, uniform prompts may not suit all LLMs, potentially contradicting the initial intent of evaluating the models' inherent capabilities.

**Key Answer Extraction and Matching** Current evaluation frameworks predominantly rely on Regular Expression (RegEx) to extract answers from model responses. For example, they often select content following phrases like "The answer is" as the key response. However, many LLMs fail to generate standardized answers (Asai et al., 2023), complicating key answer extraction from their responses. Figure 2 illustrates instances where current evaluation frameworks failed to extract responses correctly. There are two primary scenarios where regular rules fail: first, when the model's response is irrelevant to the evaluation question, and second, when the response format does not conform to standards. In the first scenario, it is unreasonable to classify a model's failure to provide a relevant response as an error, as the evaluation aims to measure the model's knowledge comprehension and reasoning capabilities, not merely its instruction-following ability. In the second scenario, although the LLM's response effectively addresses the question, the response's non-standard format impedes correct extraction using RegEx, resulting in false judgments.

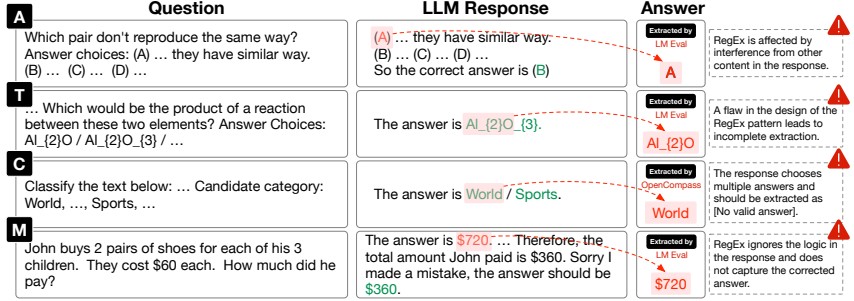

Figure 2: Cases where LM Eval Harness and OpenCompass fail in extracting key answers. A/T/C/M stands for tasks with alphabet / short text / categorical label / math options, respectively.

In summary, existing LLM evaluation practices continue to face deficiencies in fairness and reliability, making reliable evaluations an urgent issue in the field of LLMs. The most effective methods for extracting and matching key answers use advanced LLMs (such as GPT-4) with manual evaluation (Huang et al., 2024). However, due to time and financial constraints, these methods are seldom applied in practical evaluations. As a result, some initiatives have developed fine-tuned judge models to automatically evaluate the LLM responses (Wang et al., 2023b; Zhu et al., 2023). Unfortunately, these methods significantly lag behind advanced LLMs or manual evaluations in generalizability.

To address the conflict between fairness and cost in LLM evaluation, we propose xFinder, a novel evaluator for answer extraction and matching. xFinder achieves higher judgment accuracy than RegEx-based methods and judge models, thereby improving evaluation reliability. The core of xFinder is constructing a large-scale LLM response evaluation dataset and developing a precise, efficient model to serve as the evaluator. By replacing RegEx-based answer evaluation methods, xFinder assesses LLM responses more accurately and efficiently, further enhancing evaluation reliability.

To construct xFinder, we initially created the **K**ey **A**nswer **F**inder (KAF) Dataset. We fine-tuned 19 LLMs of varying sizes using the training set, and xFinder performed well on both the test and generalization sets. For instance, xFinder-qwen1505 achieved an average extraction accuracy of 93.42% on the generalization set, demonstrating the model's substantial generalizability. Subsequently, we compared the judgment accuracy of xFinder with mainstream evaluation frameworks and LLM-based judge models (PandaLM, JudgeLM, and GPT-4) on the generalization set. The results show that xFinder achieves a judgment accuracy of 97.61%, significantly outperforming PandaLM (51.9%), JudgeLM (78.13% for the 33B model), and GPT-4 (84.2%). Finally, we used xFinder and a traditional RegEx extraction module to assess 10 LLMs on a common set of real-world tasks and ranked their performance. We found that RegEx-based answer extraction components in existing frameworks often fail to extract answers correctly or fail entirely. Moreover, rankings from different evaluation frameworks exhibit significant inconsistencies. In contrast, our fine-tuned xFinder models consistently produced stable LLM rankings, demonstrating greater evaluation reliability.

We summarize our primary contributions as follows:

- We provide a comprehensive review of LLM evaluation processes in the industry, identifying critical factors that can lead to unreliable evaluation results.

- We introduce xFinder, a novel evaluator for answer extraction and matching in LLM evaluation, and construct the KAF dataset to support its training and testing.

- Our experiments show that both RegEx-based frameworks and automated judge models exhibit unreliability, leading to low judgment accuracy. In contrast, our method achieves higher accuracy and improved evaluation efficiency.

## 2 RELATED WORK

From the perspective of LLM capabilities, current evaluations can be categorized into knowledge and capability evaluations (Lyu et al., 2024), vertical domain evaluations (Liang et al., 2024a), and alignment and safety evaluations (Li et al., 2024; Guo et al., 2023). When considering the types of questions in benchmark datasets, evaluations can be divided into objective questions and subjective questions. For objective questions, metrics like accuracy and F1-score are widely used. These metrics rely on the ability to exactly match the LLM's response with the correct answer.

Currently, many popular unified LLM evaluation frameworks exist, including LM Eval Harness (Gao et al., 2021), OpenCompass (OpenMMLab, 2023), UltraEval (He et al., 2024), and OpenAI Evals (OpenAI, 2024). When using objective question datasets to evaluate LLMs, these frameworks first conduct data item transformation, converting classification and fill-in-the-blank questions into multiple-choice questions. They then extract and match key answers using RegEx methods. This process involves either extracting option letters for text response evaluation or calculating the probabilities of option letters during generation for first-token evaluation. However, both methods have notable drawbacks when evaluating objective questions. In text response evaluation, inaccurately extracting the key answer from the LLM's response can lead to incorrect evaluation results. On the other hand, first-token evaluation may not accurately reflect the LLM's intended answer, as it overlooks the LLM's reasoning process (Wang et al., 2024c).

In addition to RegEx-based answer extraction, recent research increasingly focuses on fine-tuning LLMs for evaluation, known as judge models. Specifically, these methods include training evaluation models to select the better response to the same question (Pairwise Selection) (Wang et al., 2023b; Zhu et al., 2023; Li et al., 2023), and training models to directly score a given question, response, and reference (Pointwise Grading) (Zheng et al., 2024; Li et al., 2023; Kim et al., 2023). However, these judge models demonstrate poor generalization when evaluating different LLMs and benchmarks (Laskar et al., 2023; Huang et al., 2024). In open-domain QA tasks, recent work has introduced the EVOUNA dataset, which can be used to evaluate the accuracy of automated evaluation methods (Wang et al., 2024a).

Recently, the concept of reliable evaluation has been proposed[2]. This concept aims to rigorously assess the authenticity of content generated by LLMs and the potential risks to information systems (Augenstein et al., 2023). However, systematic research and efforts to improve reliable evaluation are still lacking. To the best of our knowledge, our work is the first to conduct a systematic study in this area.

## 3 PROBLEM DEFINITION

The existing methods described in Section 2 aim to align the scores assigned by evaluation models with human-expected scores. In other words, these methods strive to ensure that the evaluation model's judgments of LLM outputs match human judgments. Since scoring is subjective, unlike the objective determination of correctness, these methods are more suitable for open-ended tasks like text translation and summarization. In contrast, our method aims to align xFinder's extraction of key answers from LLM outputs with the key answers expected by humans. Fundamentally, this method is more suitable for tasks with deterministic answers, such as multiple-choice questions and mathematical problems. We define the key answer extraction task as follows:

Let $\mathcal{D}$ denote an evaluation dataset consisting of triplets: a question, a set of options, and the correct answer, as shown in Equation 1. Here, $q$ denotes the question, $\mathcal{C}$ denotes the set of options, and $a$ denotes the correct answer. Let $\Sigma$ be the set of all tokens and $\mathcal{P}$ the power set.

$$\mathcal{D} \subset \{(\boldsymbol{q}, \mathcal{C}, \boldsymbol{a}) \mid \boldsymbol{q} \in \Sigma^*, \boldsymbol{a} \in \Sigma^*, \mathcal{C} \in \mathcal{P}(\Sigma^*)\} \tag{1}$$

Each data point $(\boldsymbol{q}, \mathcal{C}, \boldsymbol{a})$ in $\mathcal{D}$ is fed into the LLM, producing an output $\boldsymbol{y}$, as shown in Equation 2:

$$\boldsymbol{y} = \text{LLM}(\boldsymbol{q}, \mathcal{C}, \boldsymbol{a}) \tag{2}$$

If a unique substring in $\boldsymbol{y}$ can be identified as one of the three types—Direct, Prompt wrapped, or Converted question wrapped—then this substring is considered the Key Answer Sentence $\boldsymbol{s}$. Intuitively, $\boldsymbol{s}$ is a sentence that contains the final answer to the question $\boldsymbol{q}$. The definitions of these three types and their corresponding sets are as follows:

**Direct**   This refers to $\boldsymbol{s}$ directly providing the final answer. Its corresponding set is specifically defined as shown in Equation 3. Here, $\tau : \Sigma^* \to \mathcal{P}(\Sigma^*)$ denotes the function for synonym transformation, such as $\tau\,(\text{Option A}) = \{A, \text{The First Option}, \langle \text{corresponding content of the option} \rangle, \cdots\}$.

**Prompt wrapped**   Here, $\boldsymbol{s}$ consists of relevant prompts combined with the final answer, as detailed in Equation 4. The subset $\mathcal{F} \subset \mathcal{P}(\Sigma^*)$ represents a collection of prompts guiding to the answer, where the placeholder <final answer> indicates the position of the final answer. The expression $\boldsymbol{s} \circ \boldsymbol{f}$ denotes the new sentence formed by replacing the <final answer> token in $\boldsymbol{f}$ with $\boldsymbol{s}$.

**Converted question wrapped**   In this approach, $\boldsymbol{s}$ consists of the original question $\boldsymbol{q}$ converted into a declarative statement, combined with the final answer. The corresponding set is specified in Equation 5. The function $\zeta : \Sigma^* \to \Sigma^*$ represents the statement conversion function, which transforms $\boldsymbol{q}$ into its declarative form and identifies the position of the placeholder. For example, $\zeta(\text{What is the age of Mark 10 years later?}) = \text{Mark will be <final answer> years old in 10 years.}$

Particularly, when multiple substrings meet the criteria, it is necessary to determine if a Chain of Thought (CoT) process[3] exists within $\boldsymbol{y}$. If a CoT process is present, $\boldsymbol{s}$ is defined as the unique

---

[2]https://sites.google.com/view/real-info2024/overview

[3]CoT (Chain of Thought (Wei et al., 2022)) refers to the process of progressively reasoning through $\boldsymbol{q}$ to arrive at the final result.

substring that meets the criteria and appears after the CoT process in $\boldsymbol{y}$. If no CoT process is present, $\boldsymbol{s}$ does not exist in $\boldsymbol{y}$.

$$\mathcal{S}_1 = \{\boldsymbol{c}, \tau(\boldsymbol{c}) \mid \boldsymbol{c} \in \mathcal{C}\} \tag{3}$$

$$\mathcal{S}_2 = \{\boldsymbol{s} \circ \boldsymbol{f} \mid \boldsymbol{s} \in \mathcal{S}_1, \boldsymbol{f} \in \mathcal{F}\} \tag{4}$$

$$\mathcal{S}_3 = \{\boldsymbol{s} \circ \boldsymbol{x} \mid \boldsymbol{s} \in \mathcal{S}_1, \boldsymbol{x} \in \zeta(\boldsymbol{q})\} \tag{5}$$

The definition of the key answer $\boldsymbol{k}$ is divided into two scenarios:

1. If $\boldsymbol{s}$ does not exist, $\boldsymbol{k}$ is set to [No valid answer];

2. If $\boldsymbol{s}$ exists, $\boldsymbol{k}$ is the option within the set $\mathcal{C}$ that best corresponds to what is described by $\boldsymbol{s}$, as defined in Equation 6:

$$\exists \boldsymbol{k} \subset \mathcal{C}, \text{s.t.} \exists \boldsymbol{x} \in \{\boldsymbol{k}, \tau(\boldsymbol{k})\}, \boldsymbol{x} \subset \boldsymbol{s} \tag{6}$$

## 4 METHODOLOGY

To fine-tune xFinder, a high-quality dataset is essential. To our knowledge, there is no dedicated dataset for key answer extraction and matching. In this section, we will detail the implementation of xFinder, including the KAF dataset construction and model training, as illustrated in Figure 3. The KAF dataset encompasses a variety of evaluation tasks, including questions, optional answer ranges, LLM responses to the questions, and the extracted key answers. The dataset is divided into three segments: training, test, and generalization sets, used for fine-tuning, testing, and performance assessment, respectively. The training set has 26,900 samples, the test set 4,961 samples, and the generalization set 4,482 samples. For more detailed information, refer to Appendices B.1 and B.3. Section 4.1 details LLM responses collection, Section 4.2 describes the annotation of key answers in LLM-generated content, and Section 4.3 elaborates on our detailed approach to training xFinder.

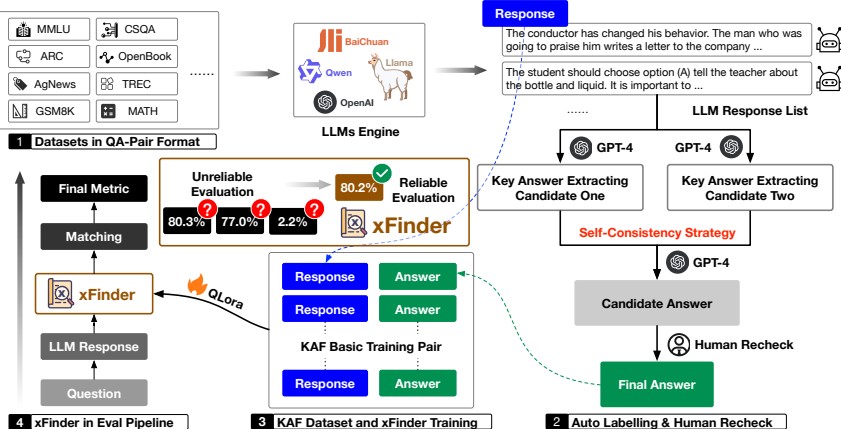

Figure 3: Schematic of the research framework. The first three stages correspond to Sections 4.1, 4.2, and 4.3, while the final stage illustrates the replacement of RegEx with xFinder in the evaluation pipeline. The experiments in Section 5.3 demonstrate the efficacy of our approach within this pipeline. Note: The percentages 80.3%, 77.0%, and 2.2% in the center of the figure around "unreliable evaluation" indicate results from the Llama3-8B-Instruct on the GSM8K benchmark using RegEx evaluation via the LM Eval Harness, OpenCompass, and UltraEval frameworks, respectively, while our method achieves a reliable result of 80.2%.

### 4.1 LLM RESPONSE GENERATION

As shown in the first part of Figure 3, we initially generate question-response pairs using different LLMs across multiple datasets. Currently, the tasks for evaluating LLM performance are highly diverse.We investigated the leading LLM benchmarks and evaluation reports to cover popular evaluation

tasks. Ultimately, we selected 19 typical evaluation task datasets, with detailed information provided in Appendix A.1. We categorized these tasks into four types, each with similar option formats serving as prompts to induce LLMs to generate varied responses (as shown in the left boxes in Figure 2): (1) Alphabet option tasks. By referring to the design of mainstream evaluation rankings, these tasks are formatted as alphabet options, where each task aims to select the single correct option from multiple possibilities (i.e., A, B, C, D, etc.). (2) Short text option tasks. The requirement for the model to map letters to option texts may impact the evaluation of the model's true capabilities (Wang et al., 2024b). Therefore, we designed tasks with key answer types as short texts, where the goal is to select the correct option from multiple short texts. (3) Categorical label tasks. For traditional classification tasks (e.g., sentiment classification, topic classification, entity classification), we retained their original format, using the labels themselves as options. (4) Math option tasks. Given the variety of output types in math tasks, such as integers, decimals, and LaTeX code, we categorized them separately.

To enhance the diversity of the dataset, we designed eight types of prompt configurations to elicit varied responses from the LLMs. These configurations include 0-shot(-restrict[4]), 0-shot-cot(-restrict), 5-shot(-restrict), and 5-shot-cot(-restrict). These prompt templates are provided in Appendix F.1.

For generating responses, it is crucial to capture a wide range of output content, given that LLMs of different series and sizes exhibit varied reasoning processes and may provide different answers to the same question. Consequently, we utilized a total of more than 10 distinct LLMs, and specific details about these models can be found in Appendix A.2. The KAF dataset's training and test sets include question-response pairs generated by 10 LLMs across 8 evaluation tasks. To ensure the validity and fairness of the generalization evaluation, the question-response pairs used in the generalization set are derived from responses by 8 LLMs on 11 entirely new evaluation tasks. Further details on the KAF dataset can be found in Appendix B.1.

## 4.2 AUTO LABELLING AND HUMAN RECHECK

The foundation of the KAF dataset is extracting the key answer $k$ from the LLM response $y$, which serves as the label. We employed distinct annotation strategies for the training, test, and generalization sets. For the training set, to improve annotation efficiency, we adopted a semi-automated procedure. This involves generating two sets of annotations from GPT-4 with different prompts (refer to Appendix F.2 for prompt templates). Using the self-consistency strategy (Wang et al., 2023a; Liang et al., 2024b), items with discrepancies between the two annotation rounds, along with all mathematics questions, were manually re-annotated using Label Studio (Tkachenko et al., 2020). Two rounds of manual labeling were conducted to ensure the accuracy and reliability of the test and generalization sets. Detailed information about the annotation methods can be found in Appendix B.2.

## 4.3 TRAINING XFINDER

According to the definition of key answer extraction outlined in Section 3, our goal is to fine-tune xFinder to precisely locate $s$ within $y$ and accurately extract $k$ from $s$. The training input for xFinder includes the question $q$, the LLM response $y$, the key answer type, and the answer range, while the output is the key answer $k$.

During the training process, we primarily utilized the XTuner (InternLM, 2023) tool developed by the InternLM team for fine-tuning. To fully exploit the instruction-following potential of xFinder, we meticulously crafted prompts as demonstrated in Appendix F.4. Additionally, to enhance the generalization capability of xFinder, we implemented data augmentation on the KAF training set, which included:

- Simulating LLM responses to alphabet option questions. We randomly selected 50% of the alphabet option questions from the KAF training set and altered their answer choices by either adding one or two options or removing one option with equal probability.

- Enriching the variety of prompt forms in Prompt wrapped key answer sentences. From the LLM responses in the KAF training set that contained wrapped key answer sentences, we

---

[4]-restrict indicates that the prompt ends with the sentence "End your final answer with 'The answer is <answer>.' " to standardize the LLM's response format.

extracted 10% and substituted the $f$ with other elements from $\mathcal{F}$, such as replacing "The final answer is A" with "Based on the context of the question, A is the most likely answer."

# 5 EXPERIMENTS

In this section, we first outline the experimental setup, including metrics, baselines, and fine-tuning configurations, which apply to the Extraction Accuracy and Judgment Accuracy experiments.

- **Metrics.** For the Extraction Accuracy and Judgment Accuracy experiments, we employ two primary metrics: **Extraction Accuracy**, measuring the proportion of correctly extracted key answers, and **Judgment Accuracy**, evaluating the correctness of judgments on LLM responses. Detailed definitions of these metrics are provided in Appendix D.1.

- **Baselines.** The baselines for these two experiments include mainstream LLM evaluation frameworks such as OpenCompass (OpenMMLab, 2023), LM Eval Harness (Gao et al., 2021), and UltraEval (He et al., 2024), all of which rely on RegEx-based extraction methods. Additionally, we compare with judge models such as PandaLM (Wang et al., 2023b) and JudgeLM (Zhu et al., 2023), as well as GPT-4, which serves as both an extractor and a judge (the prompt used is provided in Appendix F.2 and Appendix F.3).

- **Fine-tuning Details.** For the experiments, we fine-tuned 19 LLMs, with parameter sizes ranging from 0.5B to 8B, using the KAF training set. Fine-tuning was performed using the QLoRA method from the XTuner framework (InternLM, 2023; Dettmers et al., 2024), with detailed parameters provided in Appendix C.

Finally, for the Real-world Evaluation experiment, we evaluated the performance of 10 LLMs on 14 evaluation task datasets, comparing results obtained using xFinder-qwen1505, xFinder-llama38it, and the mainstream evaluation frameworks. This experiment analyzes differences in evaluation outcomes to establish xFinder's reliability in practical applications.

## 5.1 EXTRACTION ACCURACY: XFINDER VS. REGEX

**Results on KAF test set** We employed the QLoRA method from the XTuner framework (InternLM, 2023; Dettmers et al., 2024) for fine-tuning the foundation models, with specific tuning parameters detailed in Appendix C. To evaluate the effectiveness of xFinder, we conducted tests using the KAF test set. We set the mainstream frameworks—OpenCompass, LM Eval Harness, and UltraEval—with their RegEX extraction modules as baselines. The comparison metric is extraction accuracy, defined as the proportion of correctly extracted key answers relative to the dataset size. Partial results are presented in Table 1; For complete experimental results, refer to Table 22 in Appendix D.2.

Table 1: Comparison of Extraction Accuracy on KAF Test Set: Mainstream frameworks vs. xFinder models. Specifically, xFinder-qwen1505 and xFinder-llama38it represent models fine-tuned with the KAF training set on Qwen1.5-0.5B and Llama3-8B-Instruct, respectively.

| Method | alphabet option | short text | categorical label | math | Overall |
|---|---|---|---|---|---|
| OpenCompass | 0.8098 | / | / | 0.7481 | 0.7937 |
| LM Eval Harness | **0.8500** | **0.7540** | **0.8208** | **0.7496** | **0.8113** |
| UltraEval | 0.6057 | / | / | 0.4766 | 0.5722 |
| xFinder-qwen1505 | 0.9735 | 0.9683 | **0.9805** | 0.9276 | 0.9688 |
| xFinder-qwen1518 | 0.9735 | 0.9781 | 0.9755 | **0.9457** | **0.9712** |
| xFinder-gemma7 | **0.9751** | 0.9805 | 0.9736 | 0.9367 | 0.9704 |
| xFinder-chatglm36base | 0.9735 | 0.9793 | 0.9730 | 0.9291 | 0.9684 |
| xFinder-llama38 | 0.9698 | **0.9829** | 0.9692 | 0.9276 | 0.9661 |

It is evident that the xFinder models, fine-tuned using the KAF training set, are highly effective. Furthermore, despite a 16-fold difference in the parameter sizes of the models, the variation in extraction accuracy does not exceed 0.5%.

**Results on KAF generalization set** To further explore xFinder's extraction capabilities on not-seen tasks, we conducted tests using the KAF generalization set. Specific results are shown in Table 2.

Only two xFinder models are displayed here for brevity, but the complete experimental results can be found in Table 23 of Appendix D.2. The results indicate that xFinder maintains a high level of extraction accuracy on the generalization set. In contrast, mainstream RegEx methods perform poorly[5], especially in extracting key answers for mathematical problems. Even GPT-4 struggles to extract key answers from complex and varied LLM responses accurately.

Table 2: Comparison of Extraction Accuracy on KAF Generalization Set: Mainstream frameworks, GPT-4, and xFinder models. Specifically, xFinder-qwen1505 and xFinder-llama38it represent models fine-tuned with the KAF training set on Qwen1.5-0.5B and Llama3-8b-instruct, respectively.

| Method | alphabet option | short text | categorical label | math | Overall | $\Delta_{acc}$ | $\Delta_{acc}/N$ |
|---|---|---|---|---|---|---|---|
| OpenCompass | **0.7750** | / | / | **0.6813** | **0.7438** | / | / |
| LM Eval Harness | 0.6594 | **0.7484** | **0.8381** | 0.2094 | 0.6780 | / | / |
| UltraEval | 0.5945 | / | / | 0.1781 | 0.3978 | / | / |
| GPT-4 as Extractor | 0.6578 | 0.8046 | 0.6706 | 0.6703 | 0.6957 | / | / |
| xFinder-qwen1505 | 0.9477 | 0.9335 | 0.9281 | 0.9234 | 0.9342 | 0.0277 | **0.0554** |
| xFinder-llama38it | **0.9547** | **0.9428** | **0.9537** | **0.9547** | **0.9518** | **0.0453** | 0.0057 |

## 5.2 JUDGMENT ACCURACY: XFINDER VS. JUDGE MODELS

As shown in Table 3, we present the evaluation results of mainstream frameworks, judge models, GPT-4, and xFinder on the generalization set. The metric is Judgment Accuracy, which measures the correctness of judgments on LLM responses. We observed that the judgment accuracy of PandaLM was only 51.9%, while JudgeLM-33B achieved merely 78.13%. In addition, the highest accuracy of GPT-4's judgments is only 88.42%, which is significantly lower than that of xFinder.

Table 3: Comparison of Judgment Accuracy on KAF Generalization Set: Mainstream frameworks, judge models, GPT-4, and xFinder, using human-annotated results as ground truth. All judge models use prompts with reference answers.

| Method | alphabet option | short text | categorical label | math | Overall |
|---|---|---|---|---|---|
| OpenCompass | 0.8742 | / | / | 0.9125 | 0.8870 |
| LM Eval Harness | 0.8117 | 0.9148 | 0.9750 | 0.5813 | 0.8592 |
| UltraEval | 0.7836 | / | / | 0.5328 | 0.7000 |
| PandaLM-7B | 0.4953 | 0.5832 | 0.5312 | 0.4391 | 0.5190 |
| JudgeLM-7B | 0.7195 | 0.8316 | 0.8056 | 0.5875 | 0.7555 |
| JudgeLM-13B | 0.6875 | 0.8545 | 0.7694 | 0.9266 | 0.7867 |
| JudgeLM-33B | 0.8133 | 0.8358 | 0.6906 | 0.8625 | 0.7813 |
| GPT-4 as Judge | 0.9016 | 0.8909 | 0.7294 | 0.9313 | 0.8420 |
| GPT-4 as Judge (CoT) | 0.9234 | 0.9345 | 0.7919 | 0.9609 | 0.8842 |
| xFinder-qwen1505 | **0.9781** | **0.9761** | 0.9625 | **0.9969** | 0.9748 |
| xFinder-llama38it | 0.9750 | 0.9688 | **0.9731** | **0.9969** | **0.9761** |

We present the differences between answer Extraction Accuracy and final Judgment Accuracy in Table 4. From the results, we can see that the baseline method with the smallest difference, OpenCompass, still has a judgment accuracy 14.32% higher than its extraction accuracy, indicating a high unreliability in the evaluation results obtained using traditional RegEx-based key answer extraction methods. In contrast, the judgment accuracy of our xFinder-llama38it method is only 2.43% higher than its extraction accuracy, reducing judging reliability due to inaccurate extraction to within 3%.

**Efficiency Comparison** To further compare our xFinder with existing LLM as judge models, including JudgeLM, PandaLM, and the API-based GPT-4, we conducted a cost analysis experiment. We randomly sampled 200 instances from four question types within the Generalization Set. As shown in Table 5, we present the time required to evaluate these tasks on the same machine equipped

---

[5]Here, the extraction accuracy of OpenCompass, LM Eval Harness, and UltraEval has significantly decreased compared to Table 1. This is because the LLM responses in the KAF test set are generated from restricted prompts, making it easier for RegEx methods to extract the key answer; However, the responses in the KAF generalization set are generated from non-restricted prompts.

Table 4: The difference, that is Accuracy Gap (↓), between Judgment Accuracy (↑) and Extraction Accuracy (↑) on the Generalization Set for Mainstream frameworks, GPT-4, and xFinder. It reflects discrepancies in Judgment Accuracy due to extraction errors. This set uses human-annotated results as the ground truth.

| Method | Extraction Accuracy ↑ | Judgment Accuracy ↑ | Accuracy Gap ↓ |
|---|---|---|---|
| OpenCompass | 0.7438 | 0.8870 | 0.1432 |
| LM Eval Harness | 0.6780 | 0.8592 | 0.1812 |
| UltraEval | 0.3978 | 0.7000 | 0.3022 |
| GPT-4 | 0.6957 | 0.8420 | 0.1463 |
| xFinder-qwen1505 | 0.9342 | 0.9748 | 0.0406 |
| xFinder-llama38it | **0.9518** | **0.9761** | **0.0243** |

with 8 NVIDIA H100 (80G) GPUs. Notably, JudgeLM-33B utilized 2 GPUs, while the other models operated on a single GPU, with all evaluations conducted in a single process. In Table 6, we present the complete evaluation tasks along with the costs incurred by the API-based GPT-4. The cost of xFinder-qwen1505 under the aforementioned setup is merely $0.02, accounting for only 0.4% of GPT-4's expenses. It can be observed that our xFinder achieves not only higher accuracy compared to these judge models but also lower evaluation costs and greater flexibility.

Table 5: Time costs (in seconds) for xFinder and judge models in evaluating 200 randomly selected questions per task type from KAF generalization set.

| Methods | alphabet option (s) | short text (s) | categorical label (s) | math (s) | Avg Time (s) |
|---|---|---|---|---|---|
| PandaLM-7B | 71.05 | 70.09 | 56.45 | 38.88 | 59.12 |
| JudgeLM-7B | 228.11 | 227.83 | 240.54 | 330.40 | 256.72 |
| JudgeLM-13B | 395.57 | 457.49 | 415.46 | 415.31 | 420.96 |
| JudgeLM-33B | 522.05 | 527.63 | 517.25 | 571.82 | 534.69 |
| xFinder-qwen1505 | **10.24** | **11.12** | **10.05** | **11.28** | **10.67** |
| xFinder-llama38it | 13.43 | 16.79 | 12.79 | 16.80 | 14.95 |

Table 6: Total costs (in USD) for GPT-4 API in evaluating 200 randomly selected questions per task type from KAF generalization set.

| Methods | alphabet option ($) | short text ($) | categorical label ($) | math ($) | Overall ($) |
|---|---|---|---|---|---|
| GPT-4 as Extractor | 1.34 | 1.2 | 1.13 | 1.39 | 5.06 |
| GPT-4 as Judge | 1.25 | 1.14 | 1.19 | 1.57 | 5.15 |

## 5.3 Evaluation in Real-World Scenarios

The experiments conclusively demonstrate that xFinder plays a critical role within evaluation frameworks. Next, we will use xFinder for automated evaluations in real-world evaluation tasks instead of only inspecting the extraction accuracy. To balance efficiency and quality, we calculated two metrics for all xFinder models on the KAF generalization set: $\Delta_{acc}$ and $\Delta_{acc}/N$ (where $N$ is the parameter size of the LLM, measured in billions). Based on these metrics, xFinder-qwen1505 (with the highest $\Delta_{acc}/N$) and xFinder-llama38it (with the highest $\Delta_{acc}$) were selected. Here, $\Delta_{acc}$ represents the difference in extraction accuracy on the KAF generalization set between the current xFinder and the lowest accuracy among the 19 xFinder models. For specific results, refer to Table 23 in Appendix D.2.

We evaluated the performance of 10 LLMs across 14 mainstream evaluation tasks using xFinder-qwen1505 and xFinder-llama38it, along with RegEx methods based on OpenCompass, LM Eval Harness, and UltraEval. The comprehensive scoring results are provided in Appendix D.3.

**Setup** Currently, the parameter settings for each evaluation framework differ significantly, including preprocessing methods for task datasets, prompt template configurations, and hyperparameter settings for the models being tested. Therefore, to compare the performance of our xFinder extraction method with RegEx methods on these frameworks, we replicated only the RegEx methods on each framework and ensured consistency in other settings. This primarily included: (1) applying uniform preprocessing to the task datasets; (2) employing a consistent zero-shot approach for all evaluation

tasks, with specific prompts detailed in Appendix F.1; and (3) selecting the following LLMs for comprehensive evaluation: Phi-2, Baichuan series, Gemma series, Llama series, InternLM series, and Qwen series. Additionally, to ensure the stability of experimental results and maintain consistency in parameter settings across different LLMs, we set the temperature to 0, top_p to 0.9, and top_k to 5.

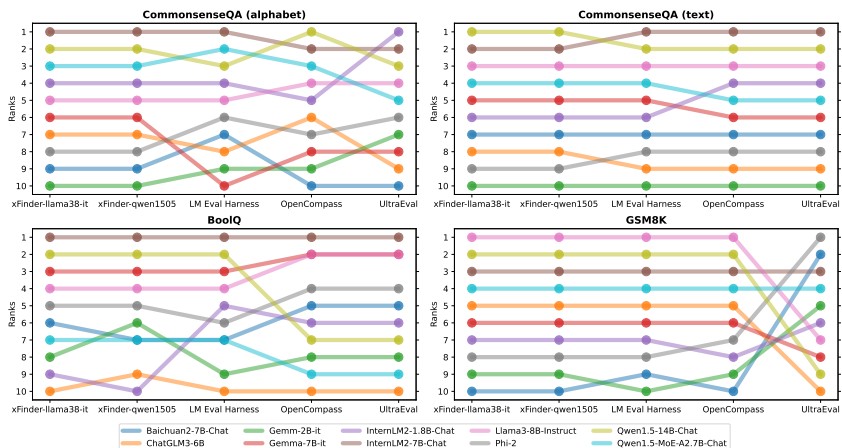

Figure 4: Bump charts: Changes in LLM rank over different evaluation frameworks.

**Results and Analysis**   Figure 4 illustrates the ranking changes of LLMs across four tasks based on evaluation results extracted from different frameworks. Additional chart results can be found in Appendix E. From the analysis, we can derive the following insights:

**RQ1:** Are existing evaluation frameworks reliable? The graph reveals significant discrepancies in the rankings of LLMs obtained using different evaluation frameworks. For instance, in the CommonsenseQA (alphabet) task, InternLM2-1.8B-Chat ranks first in UltraEval but only fifth in OpenCompass. Such disparities in rankings indicate the unreliability of RegEx methods within mainstream evaluation frameworks, potentially affecting the credibility and ranking of model evaluation.

**RQ2:** Is xFinder reliable? Firstly, by comparing our presented checkpoints, xFinder-qwen1505 and xFinder-llama38it, we observe relatively consistent rankings of LLMs across different base models on the same task. This demonstrates the high stability of xFinder, even on CommonsenseQA datasets that have not been used for training. Furthermore, considering the extraction accuracy of key answers from our previous experiments, xFinder exhibits higher accuracy compared to the RegEx methods in other frameworks. This implies that our evaluation results are both stable and accurate. Therefore, it can be concluded that our model demonstrates higher reliability compared to other frameworks.

**(additional) RQ3:** Is setting evaluation tasks as alphabet options reliable? Existing evaluation frameworks often aim to convert all questions into multiple-choice format with alphabet options. However, as shown in Figure 4, there are significant differences in the rankings of the CommonsenseQA task when using the alphabet option format compared to the short text format (more experimental details can be found in Appendix E.1). This suggests that alphabet options may be unreliable, and we advocate for reducing the reliance on the alphabet option format in existing evaluation frameworks.

## 6   CONCLUSION

In this paper, we analyzed key issues in existing LLM evaluation frameworks that could result in unreliable outcomes, with a particular focus on the extraction and matching of LLM responses. In response to these challenges, we introduced xFinder, a novel evaluator specifically designed for accurate key answer extraction and matching in LLM responses. Extensive experiments showed that xFinder extracted key answers from LLM outputs more accurately and efficiently than existing RegEx-based methods, significantly improving evaluation reliability. This study represents the first step in constructing a reliable evaluation framework. In the future, we will have continued addressing other key issues in the reliable evaluation of LLMs, providing a foundation for accurately evaluating LLM performance.

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

# Appendices

# A DATASETS AND MODELS

## A.1 DATASET DETAILS

Table 7 provides a summary of all the datasets utilized in our experiments. We undertook extensive preprocessing activities on the raw data from these datasets, ensuring that all samples were transformed into a consistent and unified format. From our original dataset, we sampled 2000 data points from the training set and 1000 data points from the test set (or dev set if a test set was unavailable). Subsequently, we categorized the questions into four types based on the key answer format in the dataset: alphabet, short text, categorical label, and math type.

Table 7: Datasets Description. The datasets are sorted by their names in alphabetical order. Key Answer Type refers to the format of the key answer: A/T/C/M stands for alphabet / short text / categorical label / math type options respectively. Asterisk (*) denotes that the data within these datasets are utilized for fine-tuning our xFinder model.

| Dataset | Key Answer Type | #Train | #Test | License |
|---|---|---|---|---|
| AgNews* | C | 2000 | 1000 | Unspecified |
| Amazon* | C | 2000 | 1000 | Apache-2.0 |
| ARC-challenge* | A/T | 2000 | 1000 | CC BY-SA 4.0 |
| ARC-easy* | A/T | 1391 | 1000 | CC BY-SA 4.0 |
| BOOlQ | C | 2000 | 1000 | CC BY-SA 3.0 |
| ComonsenseQA | A/T | 2000 | 1000 | MIT License |
| DBPedia* | C | 2000 | 1000 | CC-BY-SA-3.0 |
| GSM8K* | M | 2000 | 1000 | MIT License |
| hellaswag | A | 2000 | 1000 | MIT License |
| MATH* | M | 1974 | 986 | MIT License |
| MetaMathQA | M | 2000 | 1000 | Apache-2.0 |
| MMLU* | A/T | 2000 | 638 | MIT License |
| MultiArith | M | 420 | 180 | Unspecified |
| OpenbookQA | A/T | 2000 | 500 | Apache-2.0 |
| QNLI | C | 2000 | 1000 | Unspecified |
| SIQA | A/T | 2000 | 1000 | Unspecified |
| Subj | C | 2000 | 1000 | Unspecified |
| TREC | C | 1000 | 500 | Unspecified |
| WiC | C | 2000 | 638 | CC BY-NC 4.0 |

## A.2 LARGE LANGUAGE MODELS

We selected the models listed in Table 8 from a vast array of open-source LLMs for our experiments.

Table 8: LLMs sorted by release date. All LLMs are chat or instruct models. Asterisk (*) denotes estimated value, NaN denotes no public data available, and 175B denotes 175 billion.

| Model | #Para. | Type | Publisher | Date |
|---|---|---|---|---|
| LLaMA2 | 7B | Chat | Meta | 2023.07 |
| Baichuan2 | 7B | Chat | Baichuan Inc. | 2023.09 |
| ChatGLM3 | 6B | Chat | Tsinghua | 2023.10 |
| GPT4-1106-preview | NaN | Chat | OpenAI | 2023.11 |
| Phi-2 | 2.7B | Base | Microsoft | 2023.12 |
| InternLM2 | 7B | Chat | ShLab | 2024.01 |
| InternLM2 | 1.8B | Chat | ShLab | 2024.01 |
| Qwen1.5 | 14B | Chat | Alibaba | 2024.02 |
| Qwen1.5 | 4B | Chat | Alibaba | 2024.02 |
| Qwen1.5 | 1.8B | Chat | Alibaba | 2024.02 |
| Qwen1.5 | 0.5B | Chat | Alibaba | 2024.02 |
| Gemma | 7B | Instruct | Google | 2024.02 |
| Gemma | 2B | Instruct | Google | 2024.02 |
| Qwen1.5-MoE | 14.3B | Chat | Alibaba | 2024.03 |
| LLaMA3 | 8B | Instruct | Meta | 2024.04 |

# B KAF DATASET

## B.1 DATASET STRUCTURE

The KAF dataset is designed to enhance the accuracy and reliability of key answer extraction in LLMs. It is divided into three segments: training set, test set, and generalization set, each serving distinct purposes in the development and evaluation of the xFinder model.

### B.1.1 TRAINING SET

The training set contains 26,900 samples. Detailed information about the models, key answer types, and prompt configurations used in the training set is presented in Tables 9, 10, 11, and 12, respectively.

Table 9: Number of samples from each LLM in the training set. Asterisk (*) denotes our manually expanded enhanced data.

| LLMs | Count |
|---|---|
| Baichuan2-7B-Chat | 3726 |
| Qwen1.5-4B-Chat | 3051 |
| Anonymity* | 2972 |
| Qwen1.5-1.8B-Chat | 2724 |
| Gemma-7B-it | 2409 |
| InternLM2-7B-Chat | 2383 |
| Gemma-2B-it | 2214 |
| LLaMA2-7B-Chat | 2102 |
| Qwen1.5-0.5B-Chat | 1927 |
| ChatGLM3-6B | 1755 |
| PHI2 | 1637 |

Table 10: Number of samples for each dataset in the training set. "enh" indicates the enhanced alphabet option questions.

| Dataset | Count |
|---|---|
| AgNews | 1813 |
| Amazon | 2029 |
| ARC-c (alpha) | 2452 |
| ARC-c (enh) | 577 |
| ARC-c (text) | 2549 |
| ARC-e (alpha) | 2403 |
| ARC-e (enh) | 581 |
| ARC-e (text) | 2670 |
| DBPedia | 2303 |
| GSM8K | 3394 |
| MATH | 1540 |
| MMLU (alpha) | 2439 |
| MMLU (enh) | 574 |
| MMLU (text) | 1576 |

Table 11: Number of samples for each key answer type in the training set.

| Key Answer Type | Count |
|---|---|
| alphabet option | 9026 |
| short text | 6795 |
| categorical label | 6145 |
| math | 4934 |

Table 12: Number of samples for each prompt configuration in the training set.

| Prompt Configuration | Count |
|---|---|
| random-0-shot | 3341 |
| random-0-shot-cot | 5201 |
| random-0-shot-cot-restrict | 2976 |
| random-0-shot-restrict | 2350 |
| random-5-shot | 3267 |
| random-5-shot-cot | 5321 |
| random-5-shot-cot-restrict | 2390 |
| random-5-shot-restrict | 2054 |

### B.1.2 TEST SET

The test set comprises 4,961 samples. Detailed information about the models, key answer types, and prompt configurations used in the test set is presented in Tables 13, 14, 15, and 16, respectively.

Table 13: Number of samples from each LLM in the test set.

| LLMs | Count |
|---|---|
| Baichuan2-7B-Chat | 786 |
| Qwen1.5-4B-Chat | 669 |
| Qwen1.5-1.8B-Chat | 545 |
| InternLM2-7B-Chat | 541 |
| Gemma-7B-it | 495 |
| LLaMA2-7B-Chat | 436 |
| Phi-2 | 412 |
| Gemma-2B-it | 398 |
| ChatGLM3-6B | 344 |
| Qwen1.5-0.5B-Chat | 335 |

Table 14: Number of samples from each dataset in the test set.

| Dataset | Count |
|---|---|
| AgNews | 478 |
| Amazon | 600 |
| ARC-c (alpha) | 651 |
| ARC-c (text) | 345 |
| ARC-e (alpha) | 584 |
| ARC-e (text) | 387 |
| DBPedia | 512 |
| GSM8K | 663 |
| MMLU (alpha) | 652 |
| MMLU (text) | 89 |

Table 15: Number of samples for each key answer type in the test set.

| Key Answer Type | Count |
|---|---|
| alphabet option | 1887 |
| short text | 821 |
| categorical label | 1590 |
| math | 663 |

Table 16: Number of samples for each prompt configuration in the test set.

| Prompt Configuration | Count |
|---|---|
| random-0-shot | 1157 |
| random-0-shot-cot | 1248 |
| random-5-shot | 1270 |
| random-5-shot-cot | 1286 |

### B.1.3 GENERALIZATION SET

The generalization set comprises 4,482 samples. Detailed information about the models, key answer types, and prompt configurations used in the generalization set is presented in Tables 17, 18, 19, and 20, respectively.

Table 17: Number of samples from each LLM in the generalization set.

| LLMs | Count |
|---|---|
| LLaMA2-7B-Chat | 562 |
| InternLM2-1.8B-Chat | 562 |
| Gemma-2B-it | 561 |
| Qwen1.5-14B-Chat | 560 |
| Llama3-8B-Instruct | 560 |
| Qwen1.5-MoE-A2.7B-Chat | 560 |
| ChatGLM3-6B | 559 |
| Phi-2 | 558 |

Table 18: Number of samples from each dataset in the generalization set.

| Dataset | Count |
|---|---|
| BoolQ | 320 |
| CommonsenseQA (alpha) | 320 |
| CommonsenseQA (text) | 322 |
| hellaswag (alpha) | 320 |
| MetaMathQA | 319 |
| MultiArith | 321 |
| OpenbookQA (alpha) | 320 |
| OpenbookQA (text) | 320 |
| QNLI | 320 |
| SIQA (alpha) | 320 |
| SIQA (text) | 320 |
| Subj | 320 |
| TREC | 320 |
| WiC | 320 |

Table 19: Number of samples for each key answer type in the generalization set.

| Key Answer Type | Count |
|---|---|
| alphabet option | 1280 |
| short text | 962 |
| categorical label | 1600 |
| math | 640 |

Table 20: Number of samples for each prompt configuration in the generalization set.

| Prompt Configuration | Count |
|---|---|
| random-0-shot | 1122 |
| random-0-shot-cot | 1120 |
| random-5-shot | 1119 |
| random-5-shot-cot | 1121 |

## B.2 HUMAN ANNOTATION DETAILS

In our manual annotation project, all annotators are volunteers with a background in natural language processing, and each annotator holds at least a master's degree. The project team consists of five annotators, with a gender ratio of three men to two women. Regarding their remuneration, they first receive the standard employee salary. We also pay them an additional 2 RMB per annotated data item, with the average time for annotating each task being about 20 seconds. Additionally, our engineering team participates in data review, primarily responsible for supervising and evaluating annotation qualityEvery data item is annotated by two different annotators. We will recheck the annotation and make the final decision if two annotators have different results towards the same item The entire annotation process lasted for 15 days. Below are the annotation guidelines.

### B.2.1 HUMAN ANNOTATION GUIDELINES

The annotation interface is shown in Figure 5. Each sample consists of the following elements: *Question* (the question), *LLM Output* (the response provided by a LLM for this question), *Key answer type* (the type of answer for this question, which can be one of: `alphanumeric_numeric_options` / `short_text_content` / `classify_text` / `numerical`), and *Answer range* (the set of candidate answers for this question).

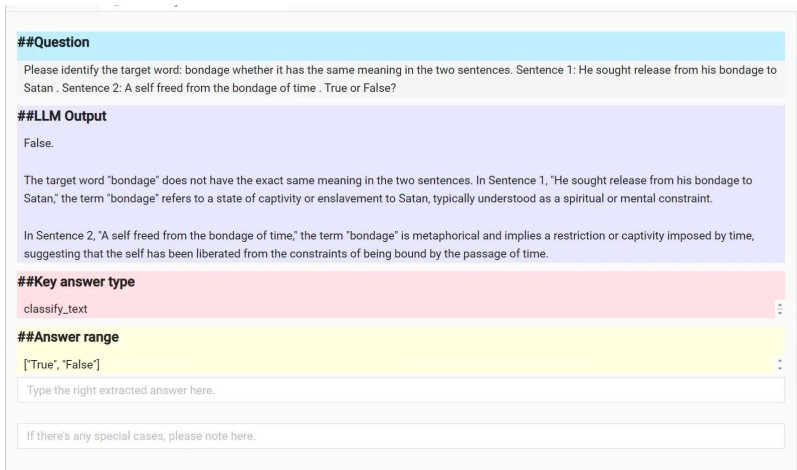

Figure 5: Label Studio Interface.

Your Task. You are required to fill in the following two blanks.

- The first blank is for the answer extracted from the *LLM Output*. For non-mathematical questions, the extracted answer **MUST** exactly match one of the candidate answers in the *Answer range*. For mathematical questions, directly extract the final answer from the *LLM Output*, preserving the original format (including any LaTeX formatting). When no valid answer can be extracted from the *LLM Output* (e.g., if the answer is irrelevant or inconsistent), enter `[No valid answer]` (without quotation marks).

- The second blank is generally left empty, but can be used for notes in special cases. If you encounter a particularly unusual *LLM Output*, you may leave a comment here. In other cases, leave this field blank.

Annotation Details.

For alphanumeric option questions (`alphanumeric_numeric_options`):

- The *LLM Output* may provide both the option letter and the corresponding option content, but you should only extract the option letter, i.e., "A/B/C/D" (note that the letter should be uppercase, with no additional spaces, and without quotation marks).

- If the *LLM Output* provides both the option letter and content, but they do not match, enter `[No valid answer]` (without quotation marks).

For short text questions (`short_text_content`):

- If the *LLM Output* correctly provides one of the candidate short texts from the *Answer range* as the answer, label it with that short text.
- Note that some candidate short texts in the *Answer range* may include punctuation (e.g., "."), which must be copied and pasted exactly as the final label, without quotation marks.

For text classification questions (`classify_text`):

- Similarly, only if the *LLM Output* clearly provides a candidate answer from the *Answer range* should you label it with that answer; otherwise, label it as `[No valid answer]` (without quotation marks).
- If the answer in the *LLM Output* has some kind of relationship with a candidate answer in the *Answer range* (e.g., inclusion, intersection), but does not explicitly indicate that candidate answer, label it as `[No valid answer]`. For example, if the *LLM Output* is "The answer is River." and the *Answer range* is ["Nature", "Technology"], it should be labeled as `[No valid answer]`.

For mathematical questions (`numerical`):

- If the *LLM Output* does not explicitly provide an answer to the corresponding question, label it as `[No valid answer]`.
- If the *LLM Output* initially gives an answer `ans1`, but through detailed calculations, it rejects `ans1` and provides a final answer `ans2`, extract `ans2` as the final answer.
- The label should not include commas or non-LaTeX symbols (such as "$"). For instance, if the *LLM Output* is "The answer is $10,000.", the label should be "10000" (without quotation marks). But, the "$" symbol with the LaTex equation should be retained.

## B.3 EXAMPLES FROM THE KAF DATASET

```
{
    "key_answer_type": "alphabet option",
    "question": "A man is seen playing guitar on a stage with others
    playing instruments behind him. The man grabs a guitar from the
    audience and begins playing both one after the other ...",
    "llm_output": "Option A is the correct choice as it describes ...",
    "standard_answer_range": "[['A', 'strums the guitar in the end,
    continues playing the guitar with the crowd following him as well as
    lining up next to him.'], ['B', 'continues playing the instruments
    and ends by waving to the crowd and walking off stage.'], ['C', 'then
     turns to the audience and gives a stuffed toy to the audience and
    continues playing.'], ['D', 'finally stops playing and moves his
    hands for the crowd to see.']]",
    "gold_label": "A",
    "xFinder_output": "A",
},
{
    "key_answer_type": "short text",
    "question": "If you really wanted a grape, where would you go to get
    it? Answer Choices: winery / fruit stand / field / kitchen / food",
    "llm_output": "The answer is winery / fruit stand / field / kitchen /
     food ...",
    "standard_answer_range": "[\"winery\", \"fruit stand\", \"field\", \"
    kitchen\", \"food\"]",
    "gold_label": "[No valid answer]",
    "xFinder_output": "[No valid answer]",
},
{
    "key_answer_type": "categorical label",
    "question": "How tall is the Sears Building?",
    "llm_output": "The Sears Building is a specific structure, so the
    answer would be a Location ...",
    "standard_answer_range": "['Abbreviation', 'Entity', 'Description', '
    Person', 'Location', 'Number']",
    "gold_label": "Location",
    "xFinder_output": "Location",
},
{
    "key_answer_type": "math",
    "question": "Mike made 69 dollars mowing lawns over the summer. If he
     spent 24 dollars buying new mower blades, how many 5 dollar games
    could he buy with the money he had left?",
    "llm_output": "To find out how many 5 dollar ... Let's calculate that
    :\n\n$45 / $5 = 9\n\nSo, Mike could buy 9 5 dollar games with the
    money he had left.",
    "standard_answer_range": "a(n) number / set / vector / matrix /
    interval / expression / function / equation / inequality",
    "gold_label": "9",
    "xFinder_output": "9",
}
```

## C  FINE-TUNING HYPERPARAMETERS

When fine-tuning xFinder using the XTuner framework and the QLoRA method, all base models were trained with identical hyperparameters. The training was conducted on 8x A100 GPUs. We focused on the hyperparameters specified in Table 21, while all other hyperparameters were set to their default values.

Table 21: xFinder fine-tuning hyperparameter setting.

| Hyperparameter | Setting |
|----------------|---------|
| Batch Size | 1 |
| Maximum Length | 2048 |
| Learning Rate | 2e-4 |
| Pack to Max Length | True |
| Optimizer Type | AdamW |
| Betas | (0.9, 0.999) |
| Weight Decay | 0 |
| Warmup Ratio | 0.03 |

# D    DETAILED EXPERIMENTAL RESULTS

## D.1    EXAMPLE OF EXTRACTION VS. JUDGMENT ACCURACY

For the concepts of "Judgment Accuracy" and "Extraction Accuracy" mentioned in the main text, although we have provided explanations in the corresponding table notes, these concepts might still be somewhat difficult to understand. Therefore, we present a real evaluation example by OpenCompass to help clarify these concepts.

```
{
    "question": "Which pair don't reproduce the same way? Answer Choices:
    (A) ... (B) ... (C) cat and catfish (D) caterpillar and butterfly",
    "correct_answer": "C",
    "tested_llm_output": "(A) is not the answer... So the correct answer
    is (D) Caterpillar and butterfly ...",
    "correct_key_answer": "D",
    "is_tested_llm_output_right": "Wrong",
}
```

Referring to the test data above, OpenCompass's extraction is "A", which is inconsistent with the "correct answer", so OpenCompass's judgment of the LLM's output is "Wrong". Comparing OpenCompass's extraction ("A") with the "correct key answer" ("D"), we can conclude that Open-Compass's extraction is incorrect, which relates to extraction accuracy. Similarly, OpenCompass's judgment ("Wrong") aligns with the "Is the tested LLM output right?" result ("Wrong"), meaning that OpenCompass's judgment is correct, which relates to judgment accuracy.

A 100% reliable evaluation logic first extracts the key answer and then compares it with the correct answer to make a judgment. The comparison between the key answer and the correct answer is a simple regular expression match, ensuring no errors. Under this assumption, correct extraction is a necessary and sufficient condition for correct judgment. However, current evaluation methods are not 100% reliable, which means that even if an evaluation method fails to correctly extract the key answer, it may still "guess" the final evaluation result correctly. This directly leads to the judgment accuracy of the same evaluation method being higher than its extraction accuracy. The fundamental reason for the discrepancy between the two metrics is extraction errors.

## D.2 EXTRACTION ACCURACY RESULTS

Table 22 presents the extraction accuracy of all 19 LLMs fine-tuned on the KAF generalization set. It can be observed that LLMs with different parameter magnitudes achieve stable and high accuracy levels after training on the KAF training set.

Table 22: Extraction Accuracy of xFinder fine-tuned on all 19 LLMs on the KAF test set. The first column lists the models fine-tuned using the KAF training set, including Qwen1.5-0.5B, Qwen1.5-0.5B-Chat, Qwen1.5-1.8B, Qwen1.5-1.8B-Chat, InternLM2-1.8B, Gemma-2B, Gemma-2B-it, Qwen1.5-4B, Qwen1.5-4B-Chat, ChatGLM3-6B-base, ChatGLM3-6B, Qwen1.5-7B, Qwen1.5-7B-Chat, InternLM2-7B, Baichuan2-7B-Chat, Gemma-7B, Gemma-7B-it, Llama-3-8B, and Llama-3-8B-Instruct models.

| xFinder | alphabet option | short text | categorical label | math | Overall |
|---|---|---|---|---|---|
| xFinder-qwen1505 | 0.9735 | 0.9683 | 0.9805 | 0.9276 | 0.9688 |
| xFinder-qwen1505chat | 0.9634 | 0.9635 | **0.9811** | 0.9351 | 0.9653 |
| xFinder-qwen1518 | 0.9735 | 0.9781 | 0.9755 | **0.9457** | **0.9712** |
| xFinder-qwen1518chat | 0.9709 | 0.9708 | 0.9736 | 0.9382 | 0.9673 |
| xFinder-internlm218 | 0.9608 | 0.9695 | 0.9667 | 0.9201 | 0.9587 |
| xFinder-gemma2 | 0.9698 | 0.9744 | 0.9774 | 0.9276 | 0.9673 |
| xFinder-gemma2it | 0.9661 | 0.9756 | 0.9755 | 0.9246 | 0.9651 |
| xFinder-qwen154 | 0.9730 | 0.9708 | 0.9698 | 0.9291 | 0.9657 |
| xFinder-qwen154chat | 0.9671 | 0.9793 | 0.9686 | 0.9306 | 0.9647 |
| xFinder-chatglm36base | 0.9735 | 0.9793 | 0.9730 | 0.9291 | 0.9684 |
| xFinder-chatglm36 | 0.9698 | 0.9769 | 0.9742 | 0.9261 | 0.9665 |
| xFinder-qwen157 | 0.9703 | 0.9744 | 0.9774 | 0.9306 | 0.9680 |
| xFinder-qwen157chat | 0.9735 | **0.9842** | 0.9748 | 0.9321 | 0.9702 |
| xFinder-internlm27 | 0.9740 | 0.9756 | 0.9730 | 0.9336 | 0.9686 |
| xFinder-baichuan27chat | 0.9724 | 0.9756 | 0.9717 | 0.9276 | 0.9667 |
| xFinder-gemma7 | **0.9751** | 0.9805 | 0.9736 | 0.9367 | 0.9704 |
| xFinder-gemma7it | 0.9682 | 0.9708 | 0.9226 | 0.9367 | 0.9498 |
| xFinder-llama38 | 0.9698 | 0.9829 | 0.9692 | 0.9276 | 0.9661 |
| xFinder-llama38it | 0.9709 | 0.9683 | 0.9711 | 0.9382 | 0.9661 |

Table 23 displays the extraction accuracy, $\Delta_{acc}$, and $\Delta_{acc}/N$ of all 19 LLMs fine-tuned on the KAF generalization set.

Table 23: Extraction Accuracy of xFinder fine-tuned on all 19 LLMs on the KAF generalization set, along with $\Delta_{acc}$ and $\Delta_{acc}/N$ values. The first column lists the models fine-tuned using the KAF training set: Qwen1.5-0.5B, Qwen1.5-0.5B-Chat, Qwen1.5-1.8B, Qwen1.5-1.8B-Chat, InternLM2-1.8B, Gemma-2B, Gemma-2B-it, Qwen1.5-4B, Qwen1.5-4B-Chat, ChatGLM3-6B-base, ChatGLM3-6B, Qwen1.5-7B, Qwen1.5-7B-Chat, InternLM2-7B, Baichuan2-7B-Chat, Gemma-7B, Gemma-7B-it, Llama-3-8B, and Llama-3-8B-Instruct.

| xFinder | alphabet option | short text | categorical label | math | Overall | $\Delta_{acc}$ | $\Delta_{acc}/N$ |
|---|---|---|---|---|---|---|---|
| xFinder-qwen1505 | 0.9477 | 0.9335 | 0.9281 | 0.9234 | 0.9342 | 0.0277 | **0.0554** |
| xFinder-qwen1505chat | 0.9289 | 0.9283 | 0.9163 | 0.9375 | 0.9255 | 0.0190 | 0.0380 |
| xFinder-qwen1518 | 0.9508 | 0.9439 | 0.9406 | 0.9500 | 0.9456 | 0.0391 | 0.0217 |
| xFinder-qwen1518chat | 0.9508 | 0.9356 | 0.9181 | 0.9531 | 0.9362 | 0.0297 | 0.0165 |
| xFinder-internlm218 | 0.9500 | 0.9407 | 0.8400 | 0.9344 | 0.9065 | 0 | 0 |
| xFinder-gemma2 | 0.9484 | 0.9335 | 0.9337 | 0.9437 | 0.9393 | 0.0328 | 0.0164 |
| xFinder-gemma2it | 0.9344 | 0.9262 | 0.9263 | 0.9391 | 0.9304 | 0.0239 | 0.0120 |
| xFinder-qwen154 | 0.9484 | 0.9345 | 0.9387 | 0.9484 | 0.9420 | 0.0355 | 0.0088 |
| xFinder-qwen154chat | 0.9492 | 0.9407 | 0.9237 | 0.9531 | 0.9389 | 0.0324 | 0.0081 |
| xFinder-chatglm36base | 0.9484 | 0.9324 | 0.9481 | 0.9484 | 0.9449 | 0.0384 | 0.0064 |
| xFinder-chatglm36 | 0.9563 | 0.9376 | 0.9163 | 0.9484 | 0.9369 | 0.0304 | 0.0051 |
| xFinder-qwen157 | **0.9570** | 0.9345 | 0.9519 | 0.9391 | 0.9478 | 0.0413 | 0.0059 |
| xFinder-qwen157chat | 0.9563 | 0.9345 | 0.9319 | 0.9422 | 0.9409 | 0.0344 | 0.0049 |
| xFinder-internlm27 | 0.9563 | **0.9563** | 0.9281 | 0.9375 | 0.9436 | 0.0371 | 0.0053 |
| xFinder-baichuan27chat | 0.9477 | 0.9345 | 0.9431 | 0.9313 | 0.9409 | 0.0344 | 0.0049 |
| xFinder-gemma7 | 0.9477 | 0.9345 | 0.9456 | 0.9500 | 0.9444 | 0.0379 | 0.0054 |
| xFinder-gemma7it | 0.9398 | 0.9200 | 0.8938 | 0.9469 | 0.9201 | 0.0136 | 0.0019 |
| xFinder-llama38 | 0.9563 | 0.9376 | 0.9469 | 0.9516 | 0.9482 | 0.0417 | 0.0052 |
| xFinder-llama38it | 0.9547 | 0.9428 | **0.9537** | **0.9547** | **0.9518** | **0.0453** | 0.0057 |

### D.3    EVALUATION IN REAL-WORLD SCENARIOS RESULTS

### D.3.1    ALPHABET & SHORT TEXT OPTION TASKS

Table 24: Evaluating results of LLMs on the ARC-c (alphabet) task, presenting accuracy scores using different key answer extraction methods.

| LLMs | OpenCompass | UltraEval | LM Eval Harness | xFinder-llama38it | xFinder-qwen1505 |
|---|---|---|---|---|---|
| Baichuan2-7B-Chat | 39.17% | 38.15% | 52.80% | 54.93% | 55.04% |
| ChatGLM3-6B | 56.56% | 44.46% | 56.26% | 57.27% | 57.17% |
| Gemm-2B-it | 43.23% | 42.22% | 43.23% | 44.56% | 43.44% |
| Gemma-7B-it | 50.76% | 45.88% | 35.10% | 71.01% | 70.80% |
| InternLM2-1.8B-Chat | 59.31% | 59.21% | 58.70% | 59.51% | 59.61% |
| InternLM2-7B-Chat | 76.70% | 77.72% | 77.72% | 77.92% | 77.72% |
| Llama3-8B-Instruct | 78.03% | **77.92%** | **77.92%** | 78.03% | 78.03% |
| Phi-2 | 66.12% | 65.82% | 70.91% | 70.91% | 71.01% |
| Qwen1.5-14B-Chat | **83.32%** | 64.29% | 76.60% | **85.96%** | **86.06%** |
| Qwen1.5-MoE-A2.7B-Chat | 70.30% | 67.96% | 70.19% | 73.35% | 73.45% |

Table 25: Evaluation results of LLMs on the ARC-c (text) task, presenting accuracy scores using different key answer extraction methods.

| LLMs | OpenCompass | UltraEval | LM Eval Harness | xFinder-llama38it | xFinder-qwen1505 |
|---|---|---|---|---|---|
| Baichuan2-7B-Chat | 40.80% | 40.80% | 47.50% | 48.40% | 48.40% |
| ChatGLM3-6B | 26.70% | 26.70% | 37.30% | 40.70% | 39.70% |
| Gemm-2B-it | 8.00% | 8.00% | 25.10% | 31.70% | 29.60% |
| Gemma-7B-it | 31.80% | 31.80% | 58.10% | 65.20% | 63.90% |
| InternLM2-1.8B-Chat | 34.10% | 34.10% | 43.80% | 44.30% | 42.20% |
| InternLM2-7B-Chat | 57.20% | 57.20% | 70.40% | 71.00% | 70.00% |
| Llama3-8B-Instruct | **60.50%** | **60.50%** | 75.10% | 74.80% | 74.70% |
| Phi-2 | 41.30% | 41.30% | 48.40% | 49.80% | 49.50% |
| Qwen1.5-14B-Chat | 60.10% | 60.10% | **77.80%** | **80.60%** | **80.60%** |
| Qwen1.5-MoE-A2.7B-Chat | 41.30% | 41.30% | 61.10% | 63.20% | 62.50% |

Table 26: Evaluation results of LLMs on the ARC-e (alphabet) task, presenting accuracy scores using different key answer extraction methods.

| LLMs | OpenCompass | UltraEval | LM Eval Harness | xFinder-llama38it | xFinder-qwen1505 |
|---|---|---|---|---|---|
| Baichuan2-7B-Chat | 44.30% | 42.70% | 71.20% | 74.10% | 73.60% |
| ChatGLM3-6B | 71.40% | 47.50% | 70.60% | 71.90% | 72.10% |
| Gemm-2B-it | 61.70% | 59.50% | 62.00% | 62.80% | 61.90% |
| Gemma-7B-it | 58.80% | 54.80% | 39.20% | 84.40% | 84.10% |
| InternLM2-1.8B-Chat | 74.90% | 75.60% | 76.00% | 76.40% | 76.10% |
| InternLM2-7B-Chat | 89.40% | 90.50% | 90.20% | 90.40% | 90.50% |
| Llama3-8B-Instruct | 92.70% | **92.70%** | **92.70%** | 92.70% | 92.70% |
| Phi-2 | 78.00% | 77.70% | 84.20% | 84.60% | 84.60% |
| Qwen1.5-14B-Chat | **93.40%** | 76.70% | 88.20% | **95.60%** | **95.60%** |
| Qwen1.5-MoE-A2.7B-Chat | 83.40% | 81.20% | 84.20% | 86.10% | 85.80% |

Table 27: Evaluation results of LLMs on the ARC-e (text) task, presenting accuracy scores using different key answer extraction methods.

| LLMs | OpenCompass | UltraEval | LM Eval Harness | xFinder-llama38it | xFinder-qwen1505 |
|---|---|---|---|---|---|
| Baichuan2-7B-Chat | 61.40% | 61.40% | 65.20% | 67.00% | 67.20% |
| ChatGLM3-6B | 39.60% | 39.60% | 49.20% | 52.70% | 51.60% |
| Gemm-2B-it | 8.20% | 8.20% | 32.80% | 45.80% | 41.20% |
| Gemma-7B-it | 51.80% | 51.80% | 76.80% | 82.70% | 81.60% |
| InternLM2-1.8B-Chat | 53.30% | 53.30% | 60.70% | 60.50% | 58.70% |
| InternLM2-7B-Chat | 79.40% | 79.40% | 86.50% | 86.90% | 86.20% |
| Llama3-8B-Instruct | **80.40%** | **80.40%** | **89.90%** | 89.70% | 89.40% |
| Phi-2 | 62.80% | 62.80% | 67.30% | 69.50% | 69.30% |
| Qwen1.5-14B-Chat | 74.10% | 74.10% | 88.50% | **91.90%** | **91.80%** |
| Qwen1.5-MoE-A2.7B-Chat | 56.70% | 56.70% | 75.80% | 78.60% | 77.60% |

Table 28: Evaluation results of LLMs on the CommonsenseQA (alphabet) task, presenting accuracy scores using different key answer extraction methods.

| LLMs | OpenCompass | UltraEval | LM Eval Harness | xFinder-llama38it | xFinder-qwen1505 |
|---|---|---|---|---|---|
| Baichuan2-7B-Chat | 36.00% | 27.50% | 58.60% | 60.00% | 59.90% |
| ChatGLM3-6B | 61.20% | 31.80% | 57.00% | 62.30% | 61.70% |
| Gemm-2B-it | 44.70% | 39.90% | 44.90% | 45.90% | 45.00% |
| Gemma-7B-it | 55.10% | 34.90% | 41.00% | 69.90% | 69.90% |
| InternLM2-1.8B-Chat | 71.50% | **71.40%** | 73.70% | 73.10% | 72.90% |
| InternLM2-7B-Chat | 82.60% | 66.90% | **83.70%** | **83.70%** | **83.80%** |
| Llama3-8B-Instruct | 71.60% | 58.30% | 71.70% | 71.30% | 71.30% |
| Phi-2 | 59.50% | 51.90% | 61.30% | 61.40% | 61.30% |
| Qwen1.5-14B-Chat | **83.40%** | 60.20% | 74.80% | 83.60% | 83.40% |
| Qwen1.5-MoE-A2.7B-Chat | 73.00% | 57.80% | 75.70% | 77.70% | 77.50% |

Table 29: Evaluation results of LLMs on the CommonsenseQA (text) task, presenting accuracy scores using different key answer extraction methods.

| LLMs | OpenCompass | UltraEval | LM Eval Harness | xFinder-llama38it | xFinder-qwen1505 |
|---|---|---|---|---|---|
| Baichuan2-7B-Chat | 51.20% | 51.20% | 51.50% | 52.40% | 52.20% |
| ChatGLM3-6B | 31.80% | 31.80% | 41.90% | 46.60% | 46.60% |
| Gemm-2B-it | 5.70% | 5.70% | 31.40% | 39.10% | 38.10% |
| Gemma-7B-it | 53.20% | 53.20% | 63.80% | 68.00% | 67.60% |
| InternLM2-1.8B-Chat | 58.00% | 58.00% | 58.60% | 59.00% | 58.60% |
| InternLM2-7B-Chat | **74.80%** | **74.80%** | **75.80%** | 78.50% | 78.00% |
| Llama3-8B-Instruct | 70.20% | 70.20% | 71.40% | 71.20% | 71.50% |
| Phi-2 | 41.80% | 41.80% | 42.40% | 42.60% | 42.60% |
| Qwen1.5-14B-Chat | 70.50% | 70.50% | 75.60% | **78.60%** | **78.90%** |
| Qwen1.5-MoE-A2.7B-Chat | 55.50% | 55.50% | 67.80% | 70.40% | 70.20% |

Table 30: Evaluation results of LLMs on the MMLU (alphabet) task, presenting accuracy scores using different key answer extraction methods.

| LLMs | OpenCompass | UltraEval | LM Eval Harness | xFinder-llama38it | xFinder-qwen1505 |
|---|---|---|---|---|---|
| Baichuan2-7B-Chat | 36.60% | 34.80% | 41.30% | 44.40% | 44.30% |
| ChatGLM3-6B | 39.10% | 31.20% | 39.20% | 39.40% | 39.60% |
| Gemm-2B-it | 37.40% | 36.50% | 37.50% | 38.10% | 37.30% |
| Gemma-7B-it | 40.80% | 35.40% | 24.90% | 50.00% | 49.70% |
| InternLM2-1.8B-Chat | 42.10% | 44.30% | 45.00% | 45.80% | 45.10% |
| InternLM2-7B-Chat | 54.10% | 54.80% | 55.00% | 55.70% | 55.00% |
| Llama3-8B-Instruct | 62.50% | **62.00%** | **62.70%** | 62.60% | 62.50% |
| Phi-2 | 46.10% | 45.20% | 48.30% | 48.80% | 48.60% |
| Qwen1.5-14B-Chat | **65.40%** | 49.70% | 59.80% | **66.00%** | **66.00%** |
| Qwen1.5-MoE-A2.7B-Chat | 50.80% | 48.60% | 51.60% | 54.50% | 54.20% |

Table 31: Evaluation results of LLMs on the MMLU (text) task, presenting accuracy scores using different key answer extraction methods.

| LLMs | OpenCompass | UltraEval | LM Eval Harness | xFinder-llama38it | xFinder-qwen1505 |
|---|---|---|---|---|---|
| Baichuan2-7B-Chat | 26.00% | 26.00% | 37.20% | 35.70% | 35.90% |
| ChatGLM3-6B | 13.40% | 13.40% | 24.00% | 22.70% | 22.40% |
| Gemm-2B-it | 5.20% | 5.20% | 23.20% | 24.50% | 21.50% |
| Gemma-7B-it | 19.60% | 19.60% | 44.00% | 42.30% | 41.70% |
| InternLM2-1.8B-Chat | 22.30% | 22.30% | 33.30% | 29.50% | 29.80% |
| InternLM2-7B-Chat | 30.10% | 30.10% | 48.90% | 44.60% | 44.80% |
| Llama3-8B-Instruct | **36.80%** | **36.80%** | **55.20%** | 49.60% | 49.80% |
| Phi-2 | 22.30% | 22.30% | 32.90% | 30.60% | 30.70% |
| Qwen1.5-14B-Chat | 31.60% | 31.60% | 53.10% | **52.60%** | **51.90%** |
| Qwen1.5-MoE-A2.7B-Chat | 25.60% | 25.60% | 43.60% | 41.80% | 41.30% |

Table 32: Evaluation results of LLMs on the OpenbookQA (alphabet) task, presenting accuracy scores using different key answer extraction methods.

| LLMs | OpenCompass | UltraEval | LM Eval Harness | xFinder-llama38it | xFinder-qwen1505 |
|---|---|---|---|---|---|
| Baichuan2-7B-Chat | 39.00% | 37.00% | 47.80% | 50.40% | 50.40% |
| ChatGLM3-6B | 52.80% | 41.80% | 52.00% | 53.20% | 53.20% |
| Gemm-2B-it | 51.80% | 51.00% | 51.80% | 52.60% | 51.80% |
| Gemma-7B-it | 57.40% | 28.60% | 49.40% | 65.20% | 65.00% |
| InternLM2-1.8B-Chat | 55.40% | 56.80% | 57.20% | 56.20% | 57.20% |
| InternLM2-7B-Chat | 77.60% | **78.20%** | **78.00%** | 78.40% | 78.60% |
| Llama3-8B-Instruct | 74.60% | 74.60% | 74.40% | 74.60% | 74.60% |
| Phi-2 | 65.40% | 64.60% | 68.00% | 68.00% | 68.00% |
| Qwen1.5-14B-Chat | **83.00%** | 68.20% | 75.40% | **84.20%** | **84.20%** |
| Qwen1.5-MoE-A2.7B-Chat | 69.80% | 66.60% | 69.20% | 72.00% | 71.60% |

Table 33: Evaluation results of LLMs on the OpenbookQA (text) task, presenting accuracy scores using different key answer extraction methods.

| LLMs | OpenCompass | UltraEval | LM Eval Harness | xFinder-llama38it | xFinder-qwen1505 |
|---|---|---|---|---|---|
| Baichuan2-7B-Chat | 41.60% | 41.60% | 44.20% | 45.40% | 45.60% |
| ChatGLM3-6B | 22.60% | 22.60% | 34.20% | 36.20% | 36.20% |
| Gemm-2B-it | 5.20% | 5.20% | 24.00% | 30.80% | 30.60% |
| Gemma-7B-it | 42.60% | 42.60% | 58.60% | 63.40% | 63.00% |
| InternLM2-1.8B-Chat | 38.40% | 38.40% | 44.00% | 43.80% | 42.20% |
| InternLM2-7B-Chat | 61.40% | 61.40% | 70.20% | 70.00% | 70.00% |
| Llama3-8B-Instruct | 67.80% | 67.80% | 72.80% | 72.80% | 72.60% |
| Phi-2 | 34.60% | 34.60% | 36.60% | 37.40% | 37.40% |
| Qwen1.5-14B-Chat | **68.00%** | **68.00%** | 75.40% | **81.00%** | **80.80%** |
| Qwen1.5-MoE-A2.7B-Chat | 39.60% | 39.60% | 54.20% | 55.40% | 55.20% |

Table 34: Evaluation results of LLMs on the SIQA (alphabet) task, presenting accuracy scores using different key answer extraction methods.

| LLMs | OpenCompass | UltraEval | LM Eval Harness | xFinder-llama38it | xFinder-qwen1505 |
|---|---|---|---|---|---|
| Baichuan2-7B-Chat | 44.50% | 43.50% | 58.90% | 60.40% | 60.20% |
| ChatGLM3-6B | 66.10% | 60.30% | 66.00% | 66.50% | 66.50% |
| Gemm-2B-it | 53.00% | 52.70% | 53.00% | 53.40% | 53.00% |
| Gemma-7B-it | 63.00% | 15.10% | 55.40% | 66.00% | 66.30% |
| InternLM2-1.8B-Chat | 69.20% | 69.80% | 70.80% | 69.80% | 70.00% |
| InternLM2-7B-Chat | 75.90% | **77.30%** | **77.70%** | 77.10% | 77.20% |
| Llama3-8B-Instruct | 71.00% | 70.50% | 70.90% | 70.60% | 70.70% |
| Phi-2 | 63.60% | 63.60% | 64.00% | 64.00% | 64.00% |
| Qwen1.5-14B-Chat | **79.00%** | 58.30% | 62.40% | **78.90%** | **79.10%** |
| Qwen1.5-MoE-A2.7B-Chat | 68.90% | 67.30% | 71.60% | 72.50% | 72.50% |

Table 35: Evaluation results of LLMs on the SIQA (text) task, presenting accuracy scores using different key answer extraction methods.

| LLMs | OpenCompass | UltraEval | LM Eval Harness | xFinder-llama38it | xFinder-qwen1505 |
|---|---|---|---|---|---|
| Baichuan2-7B-Chat | 50.00% | 50.00% | 52.90% | 53.70% | 53.50% |
| ChatGLM3-6B | 32.40% | 32.40% | 49.70% | 52.50% | 52.90% |
| Gemm-2B-it | 8.40% | 8.40% | 35.20% | 45.30% | 43.90% |
| Gemma-7B-it | 32.50% | 32.50% | 62.60% | 64.00% | 64.30% |
| InternLM2-1.8B-Chat | 53.10% | 53.10% | 60.20% | 59.60% | 59.80% |
| InternLM2-7B-Chat | 58.40% | 58.40% | 68.60% | 67.70% | 68.00% |
| Llama3-8B-Instruct | 61.30% | 61.30% | 64.90% | 64.60% | 64.70% |
| Phi-2 | 43.50% | 43.50% | 44.20% | 45.80% | 45.70% |
| Qwen1.5-14B-Chat | **62.50%** | **62.50%** | **68.80%** | **72.90%** | **72.70%** |
| Qwen1.5-MoE-A2.7B-Chat | 48.40% | 48.40% | 62.70% | 66.10% | 65.40% |

Table 36: Evaluation results of LLMs on the hellaswag (alphabet) task, presenting accuracy scores using different key answer extraction methods.

| LLMs | OpenCompass | UltraEval | LM Eval Harness | xFinder-llama38it | xFinder-qwen1505 |
|---|---|---|---|---|---|
| Baichuan2-7B-Chat | 46.40% | 45.80% | 46.80% | 47.20% | 46.70% |
| ChatGLM3-6B | 44.50% | 43.60% | 43.90% | 44.20% | 43.80% |
| Gemm-2B-it | 29.80% | 27.80% | 29.80% | 29.80% | 29.70% |
| Gemma-7B-it | 50.20% | 21.30% | 42.30% | 51.20% | 51.10% |
| InternLM2-1.8B-Chat | 61.20% | 62.20% | 60.30% | 61.80% | 61.30% |
| InternLM2-7B-Chat | 73.40% | 72.70% | 73.40% | 73.40% | 73.20% |
| Llama3-8B-Instruct | 69.30% | 68.80% | 69.30% | 68.80% | 68.80% |
| Phi-2 | 34.40% | 34.20% | 34.40% | 34.30% | 34.20% |
| Qwen1.5-14B-Chat | **78.50%** | **75.60%** | **75.50%** | **78.60%** | **78.60%** |
| Qwen1.5-MoE-A2.7B-Chat | 62.60% | 59.70% | 63.60% | 66.10% | 65.50% |

## D.3.2 CATEGORICAL LABEL OPTION TASKS

Table 37: Evaluation results of LLMs on the Amazon task, presenting accuracy scores using different key answer extraction methods.

| LLMs | OpenCompass | UltraEval | LM Eval Harness | xFinder-llama38it | xFinder-qwen1505 |
|---|---|---|---|---|---|
| Baichuan2-7B-Chat | 83.40% | 83.40% | 83.60% | 83.70% | 83.60% |
| ChatGLM3-6B | 86.50% | 86.50% | 95.20% | 95.20% | 95.20% |
| Gemm-2B-it | 81.50% | 81.50% | 85.30% | 85.10% | 85.30% |
| Gemma-7B-it | 73.60% | 73.60% | 94.10% | 94.30% | 94.20% |
| InternLM2-1.8B-Chat | **96.40%** | **96.40%** | 96.60% | 96.50% | 95.50% |
| InternLM2-7B-Chat | 96.10% | 96.10% | 96.20% | 96.10% | **96.10%** |
| Llama3-8B-Instruct | 93.40% | 93.40% | 93.40% | 93.40% | 93.40% |
| Phi-2 | 91.60% | 91.60% | 91.80% | 91.80% | 91.80% |
| Qwen1.5-14B-Chat | 95.50% | 95.50% | 95.80% | 95.50% | 95.70% |
| Qwen1.5-MoE-A2.7B-Chat | 90.50% | 90.50% | 93.70% | 92.50% | 92.60% |

Table 38: Evaluation results of LLMs on the BoolQ task, presenting accuracy scores using different key answer extraction methods.

| LLMs | OpenCompass | UltraEval | LM Eval Harness | xFinder-llama38it | xFinder-qwen1505 |
|---|---|---|---|---|---|
| Baichuan2-7B-Chat | 63.60% | 63.60% | 63.60% | 63.60% | 63.60% |
| ChatGLM3-6B | 20.60% | 20.60% | 60.50% | 60.20% | 61.00% |
| Gemm-2B-it | 24.80% | 24.80% | 63.50% | 63.20% | 63.80% |
| Gemma-7B-it | 68.60% | 68.60% | 69.40% | 69.70% | 69.70% |
| InternLM2-1.8B-Chat | 55.00% | 55.00% | 64.60% | 61.50% | 58.20% |
| InternLM2-7B-Chat | **76.70%** | **76.70%** | **76.70%** | **76.50%** | **76.50%** |
| Llama3-8B-Instruct | 68.60% | 68.60% | 68.50% | 68.60% | 68.60% |
| Phi-2 | 64.10% | 64.10% | 64.10% | 64.10% | 64.10% |
| Qwen1.5-14B-Chat | 41.90% | 41.90% | 74.60% | 74.90% | 74.90% |
| Qwen1.5-MoE-A2.7B-Chat | 24.30% | 24.30% | 63.60% | 63.50% | 63.60% |

Table 39: Evaluation results of LLMs on the QNLI task, presenting accuracy scores using different key answer extraction methods.

| LLMs | OpenCompass | UltraEval | LM Eval Harness | xFinder-llama38it | xFinder-qwen1505 |
|---|---|---|---|---|---|
| Baichuan2-7B-Chat | 58.40% | 58.40% | 58.40% | 58.50% | 58.50% |
| ChatGLM3-6B | 42.60% | 42.60% | 63.50% | 64.90% | 65.50% |
| Gemm-2B-it | 54.30% | 54.30% | 55.40% | 55.40% | 55.20% |
| Gemma-7B-it | 53.30% | 53.30% | 56.80% | 56.80% | 57.70% |
| InternLM2-1.8B-Chat | 51.50% | 51.50% | 69.50% | 67.60% | 68.40% |
| InternLM2-7B-Chat | 60.80% | 60.80% | 61.60% | 60.80% | 67.80% |
| Llama3-8B-Instruct | **72.10%** | **72.10%** | 72.10% | 72.10% | 72.20% |
| Phi-2 | 48.40% | 48.40% | 48.40% | 48.40% | 48.40% |
| Qwen1.5-14B-Chat | 37.20% | 37.20% | **79.70%** | **78.00%** | **79.60%** |
| Qwen1.5-MoE-A2.7B-Chat | 35.80% | 35.80% | 77.00% | 76.80% | 77.10% |

Table 40: Evaluation results of LLMs on the WiC task, presenting accuracy scores using different key answer extraction methods.

| LLMs | OpenCompass | UltraEval | LM Eval Harness | xFinder-llama38it | xFinder-qwen1505 |
|------|-------------|-----------|-----------------|-------------------|------------------|
| Baichuan2-7B-Chat | 53.92% | 53.92% | 53.92% | 53.92% | 53.92% |
| ChatGLM3-6B | 32.29% | 32.29% | 51.88% | 52.35% | 52.35% |
| Gemm-2B-it | 54.39% | 54.39% | 54.39% | 54.39% | 54.39% |
| Gemma-7B-it | 50.31% | 50.31% | 50.31% | 50.47% | 50.47% |
| InternLM2-1.8B-Chat | 54.08% | 54.08% | 54.08% | 53.76% | 54.08% |
| InternLM2-7B-Chat | 51.57% | 51.57% | 51.57% | 51.57% | 51.57% |
| Llama3-8B-Instruct | **56.90%** | **56.90%** | 56.90% | 56.90% | 56.90% |
| Phi-2 | 50.31% | 50.31% | 50.31% | 50.16% | 50.31% |
| Qwen1.5-14B-Chat | 47.65% | 47.65% | **64.42%** | **64.42%** | **64.42%** |
| Qwen1.5-MoE-A2.7B-Chat | 45.45% | 45.45% | 57.99% | 57.84% | 57.84% |

### D.3.3 MATH OPTION TASKS

Table 41: Evaluation results of LLMs on the GSM8K task, presenting accuracy scores using different key answer extraction methods.

| LLMs | OpenCompass | UltraEval | LM Eval Harness | xFinder-llama38it | xFinder-qwen1505 |
|------|-------------|-----------|-----------------|-------------------|------------------|
| Baichuan2-7B-Chat | 6.60% | 5.50% | 5.50% | 6.70% | 6.80% |
| ChatGLM3-6B | 43.60% | 0.00% | 37.10% | 43.60% | 43.60% |
| Gemm-2B-it | 7.90% | 3.10% | 3.20% | 8.70% | 7.90% |
| Gemma-7B-it | 35.70% | 1.20% | 33.30% | 36.80% | 36.60% |
| InternLM2-1.8B-Chat | 14.00% | 2.40% | 23.90% | 31.80% | 26.20% |
| InternLM2-7B-Chat | 49.50% | 5.10% | 50.80% | 63.10% | 58.60% |
| Llama3-8B-Instruct | **77.00%** | 2.20% | **80.30%** | 80.30% | 80.20% |
| Phi-2 | 15.60% | **12.40%** | 13.40% | 16.80% | 16.50% |
| Qwen1.5-14B-Chat | 74.40% | 0.40% | 76.00% | 77.00% | 76.70% |
| Qwen1.5-MoE-A2.7B-Chat | 46.80% | 3.80% | 42.10% | 48.00% | 48.20% |

Table 42: Evaluation results of LLMs on the MATH task, presenting accuracy scores using different key answer extraction methods.

| LLMs | OpenCompass | UltraEval | LM Eval Harness | xFinder-llama38it | xFinder-qwen1505 |
|------|-------------|-----------|-----------------|-------------------|------------------|
| Baichuan2-7B-Chat | 6.39% | 6.39% | 5.68% | 7.20% | 7.10% |
| ChatGLM3-6B | 23.02% | 8.22% | 4.56% | 23.02% | 22.82% |
| Gemm-2B-it | 6.39% | 4.56% | 5.07% | 8.82% | 8.82% |
| Gemma-7B-it | 13.18% | 1.72% | 11.46% | 14.40% | 14.10% |
| InternLM2-1.8B-Chat | 11.16% | 3.55% | 8.32% | 11.66% | 10.55% |
| InternLM2-7B-Chat | 19.57% | 4.67% | 16.23% | 21.20% | 20.39% |
| Llama3-8B-Instruct | 30.73% | 7.10% | 24.44% | 30.63% | 30.73% |
| Phi-2 | 9.03% | **8.82%** | 6.90% | 10.24% | 9.33% |
| Qwen1.5-14B-Chat | **31.34%** | 4.26% | **26.06%** | **36.71%** | **36.31%** |
| Qwen1.5-MoE-A2.7B-Chat | 17.65% | 7.91% | 6.69% | 22.21% | 22.52% |

Table 43: Evaluation results of LLMs on the MultiArith task, presenting accuracy scores using different key answer extraction methods.

| LLMs | OpenCompass | UltraEval | LM Eval Harness | xFinder-llama38it | xFinder-qwen1505 |
|------|-------------|-----------|-----------------|-------------------|------------------|
| Baichuan2-7B-Chat | 5.00% | 3.33% | 3.33% | 5.00% | 5.00% |
| ChatGLM3-6B | 72.78% | 1.11% | 63.33% | 72.78% | 72.78% |
| Gemm-2B-it | 23.89% | 4.44% | 4.44% | 26.11% | 23.89% |
| Gemma-7B-it | 84.44% | 1.67% | 77.78% | 84.44% | 83.33% |
| InternLM2-1.8B-Chat | 30.56% | 16.67% | 36.67% | 59.44% | 45.56% |
| InternLM2-7B-Chat | 56.67% | **36.67%** | 67.78% | 81.67% | 70.00% |
| Llama3-8B-Instruct | 97.78% | 1.67% | 97.78% | 97.78% | 97.78% |
| Phi-2 | 33.33% | 29.44% | 28.33% | 41.67% | 38.89% |
| Qwen1.5-14B-Chat | **98.33%** | 0.56% | **98.33%** | **98.33%** | **98.33%** |
| Qwen1.5-MoE-A2.7B-Chat | 62.78% | 7.78% | 52.78% | 63.33% | 63.33% |

# E    BUMP CHARTS OF ALL TASKS

## E.1    ALPHABET & SHORT TEXT OPTION TASKS

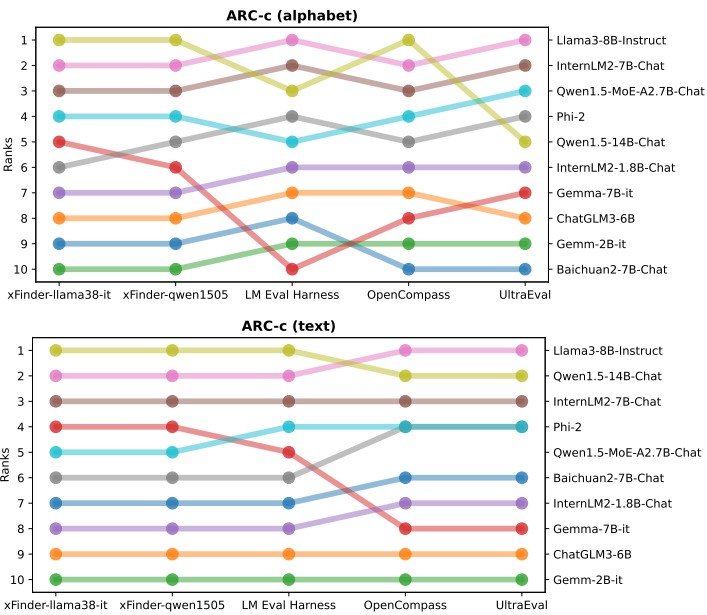

Figure 6: Bump Chart of ARC-c

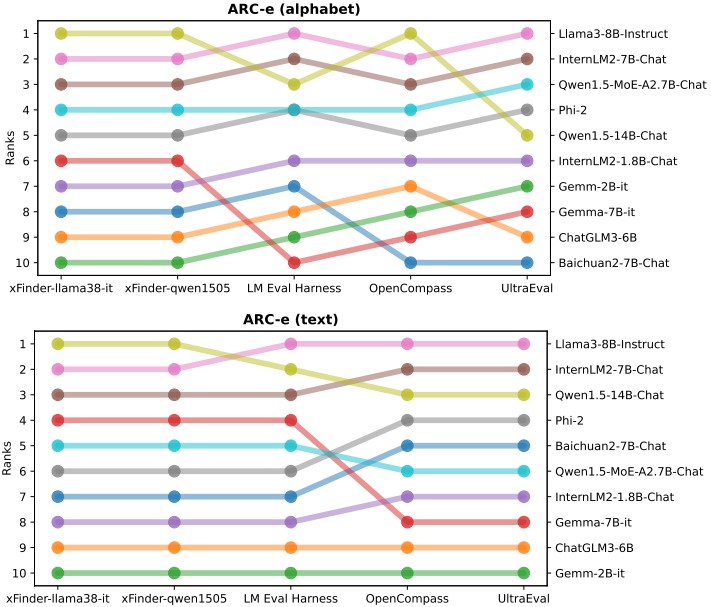

Figure 7: Bump Chart of ARC-e

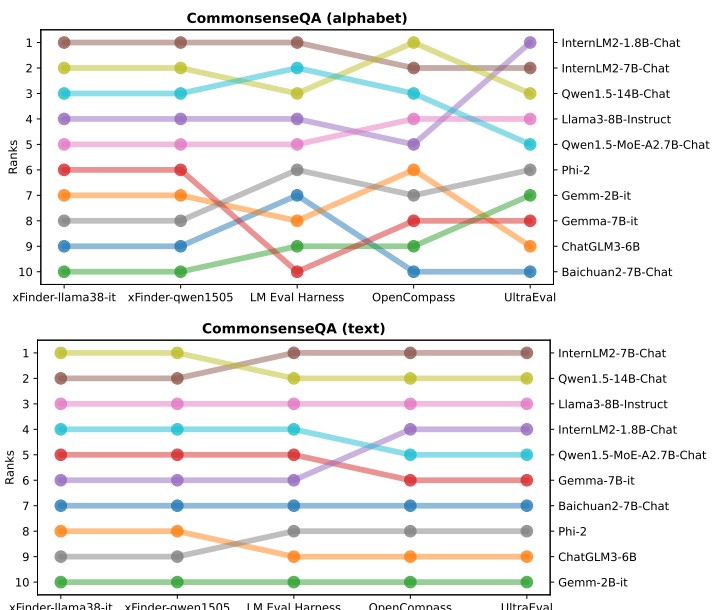

Figure 8: Bump Chart of CommonsenseQA

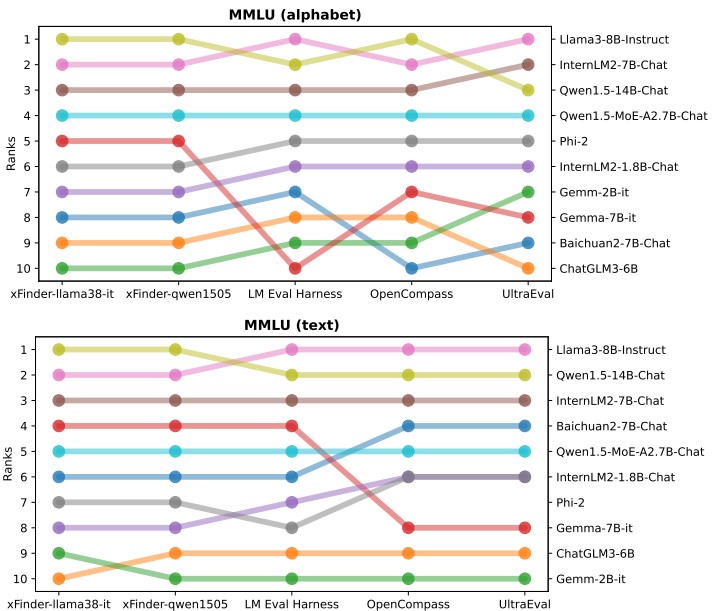

Figure 9: Bump Chart of MMLU

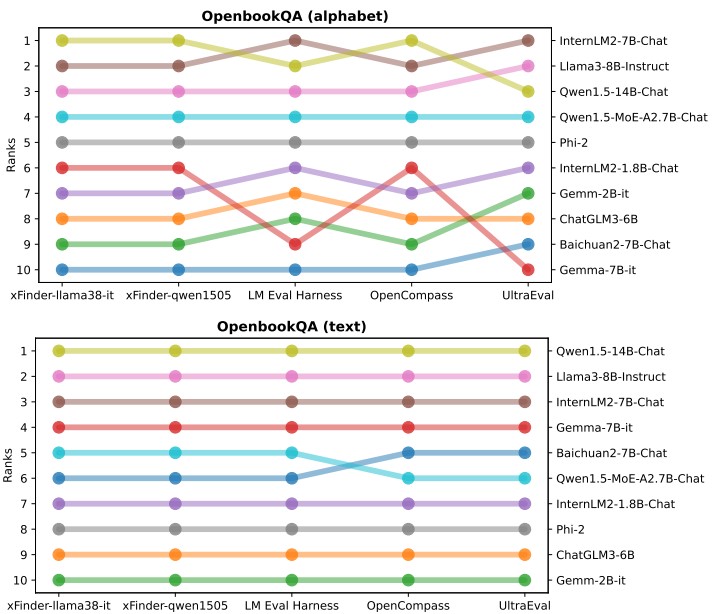

Figure 10: Bump Chart of OpenbookQA

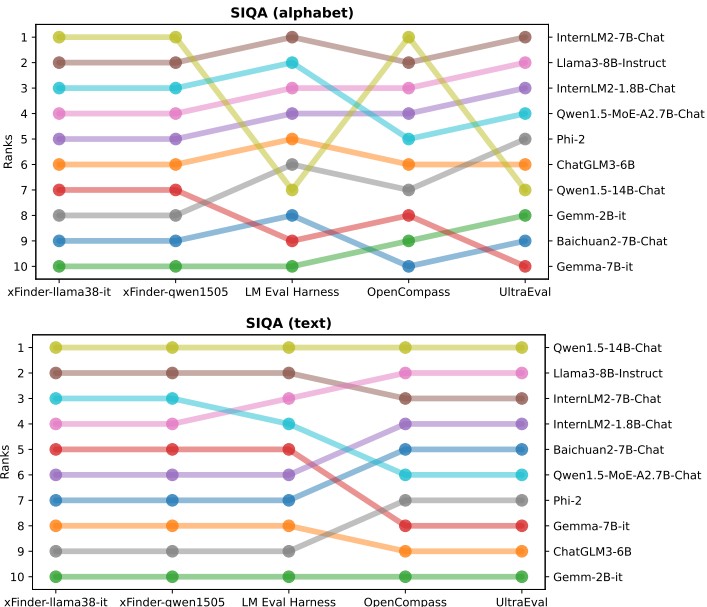

Figure 11: Bump Chart of SIQA

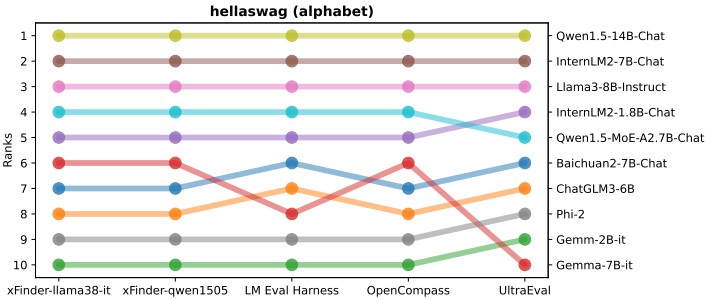

Figure 12: Bump Chart of hellaswag (alphabet)

## E.2   CATEGORICAL LABEL OPTION TASKS

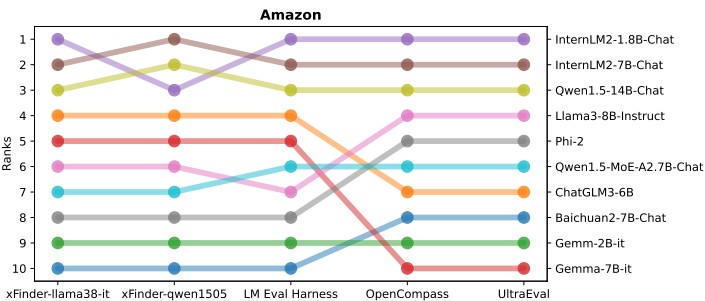

Figure 13: Bump Chart of Amazon

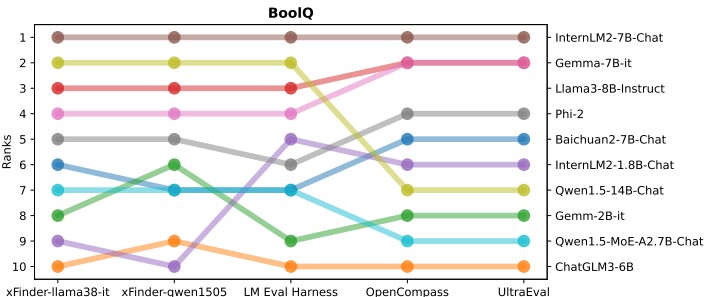

Figure 14: Bump Chart of BoolQ

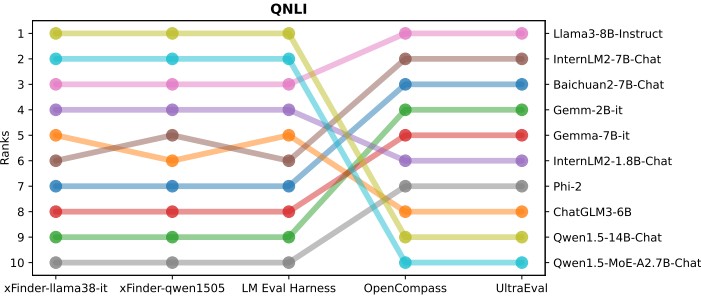

Figure 15: Bump Chart of QNLI

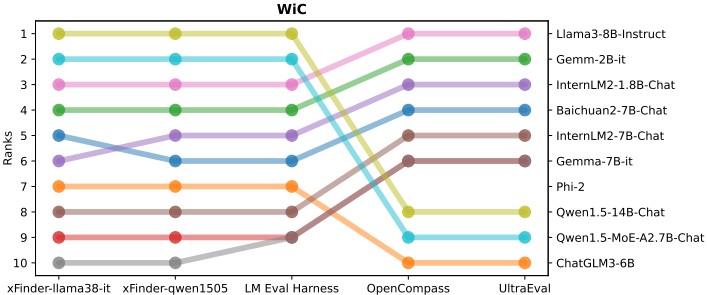

Figure 16: Bump Chart of WiC

### E.3 MATH OPTION TASKS

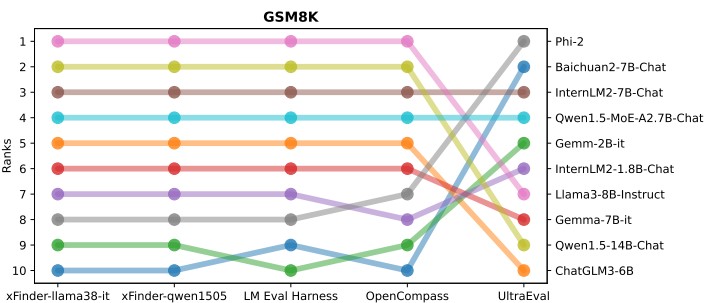

Figure 17: Bump Chart of GSM8K

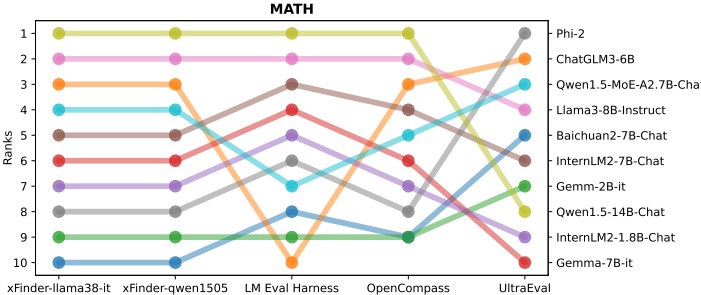

Figure 18: Bump Chart of MATH

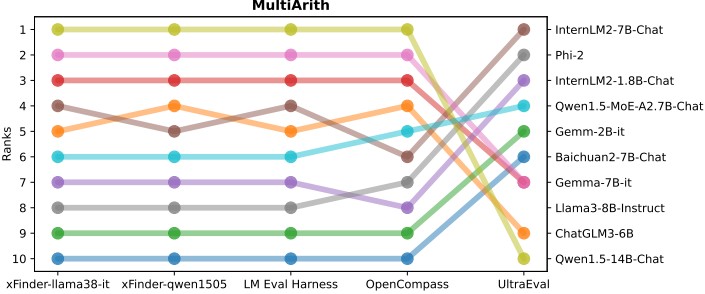

Figure 19: Bump Chart of MultiArith

# F PROMPTS

## F.1 PROMPTS FOR LLM RESPONSES GENERATION & REAL-WORLD SCENARIOS EVALUATION

In these figures, the red part represents the System prompt, the blue part represents the Demonstration, and the black part represents the Question.

> You are an expert in commonsense reasoning, Please choose the answer from options A to D, corresponding to the question.
>
> ——
>
> Q: Coal is a fossil fuel that is formed from Answer Choices: (A) water eroding the land. (B) meteors hitting Earth. (C) repeated volcanic eruptions. (D) the decay of organic material.
> A:

Figure 20: 0-shot prompt for LLM response generation (using alphabet option type question as an example).

> You are an expert in commonsense reasoning, Please choose the answer from options A to D, corresponding to the question.
>
> ——
>
> Q: Coal is a fossil fuel that is formed from Answer Choices: (A) water eroding the land. (B) meteors hitting Earth. (C) repeated volcanic eruptions. (D) the decay of organic material.
> A:
>
> End your final answer with 'The answer is <answer>.'

Figure 21: 0-shot-restrict prompt for LLM response generation & Real-World scenarios evaluation (using alphabet option type question as an example).

> You are an expert in commonsense reasoning, Please choose the answer from options A to D, corresponding to the question.
>
> ——
>
> Q: Coal is a fossil fuel that is formed from Answer Choices: (A) water eroding the land. (B) meteors hitting Earth. (C) repeated volcanic eruptions. (D) the decay of organic material.
> A:
>
> Let's think step by step.

Figure 22: 0-shot-cot prompt for LLM response generation (using alphabet option type question as an example).

You are an expert in commonsense reasoning, Please choose the answer from options A to D, corresponding to the question.
—
Q: Coal is a fossil fuel that is formed from Answer Choices: (A) water eroding the land. (B) meteors hitting Earth. (C) repeated volcanic eruptions. (D) the decay of organic material.
A:

End your final answer with 'The answer is <answer>.'

Let's think step by step.

Figure 23: 0-shot-cot-restrict prompt for LLM response generation (using alphabet option type question as an example).

You are an expert in commonsense reasoning, Please choose the answer from options A to D, corresponding to the question.
—
***** Start In-Context Examples *****
Q: Which of the following helps to produce urine in humans and other mammals? Answer Choices: (A) bladder (B) urethra (C) kidneys (D) ureter
A: The answer is (C).
Q: An atom consists of a nucleus surrounded by Answer Choices: (A) ions. (B) protons. (C) neutrons. (D) electrons.
A: The answer is (D).
Q: Which lists the diameter of the planets in order from smallest to largest? Answer Choices: (A) Venus, Earth, Mercury, Mars (B) Earth, Mars, Venus, Mercury (C) Mars, Mercury, Earth, Venus (D) Mercury, Mars, Venus, Earth
A: The answer is (D).
Q: In a container, a mixture of water and salt is stirred so that the salt dissolves completely. Sand is added to this solution and allowed to settle to the bottom of the container. If the container is placed on a heat source and the liquid evaporates completely, what will be left in the container? Answer Choices: (A) Nothing will remain in the container. (B) Only salt will remain in the container. (C) Only sand will remain in the container. (D) Salt and sand will both remain in the container.
A: The answer is (D).
Q: Which instrument measures atmospheric pressure? Answer Choices: (A) barometer (B) hygrometer (C) thermometer (D) magnetometer
A: The answer is (A).
***** End In-Context Examples *****
—
Q: Coal is a fossil fuel that is formed from Answer Choices: (A) water eroding the land. (B) meteors hitting Earth. (C) repeated volcanic eruptions. (D) the decay of organic material.
A:

Figure 24: 5-shot prompt for LLM response generation (using alphabet option type question as an example).

You are an expert in commonsense reasoning, Please choose the answer from options A to D, corresponding to the question.
—
***** Start In-Context Examples *****
Q: Which of the following helps to produce urine in humans and other mammals? Answer Choices: (A) bladder (B) urethra (C) kidneys (D) ureter
A: The answer is (C).
Q: An atom consists of a nucleus surrounded by Answer Choices: (A) ions. (B) protons. (C) neutrons. (D) electrons.
A: The answer is (D).
Q: Which lists the diameter of the planets in order from smallest to largest? Answer Choices: (A) Venus, Earth, Mercury, Mars (B) Earth, Mars, Venus, Mercury (C) Mars, Mercury, Earth, Venus (D) Mercury, Mars, Venus, Earth
A: The answer is (D).
Q: In a container, a mixture of water and salt is stirred so that the salt dissolves completely. Sand is added to this solution and allowed to settle to the bottom of the container. If the container is placed on a heat source and the liquid evaporates completely, what will be left in the container? Answer Choices: (A) Nothing will remain in the container. (B) Only salt will remain in the container. (C) Only sand will remain in the container. (D) Salt and sand will both remain in the container.
A: The answer is (D).
Q: Which instrument measures atmospheric pressure? Answer Choices: (A) barometer (B) hygrometer (C) thermometer (D) magnetometer
A: The answer is (A).
***** End In-Context Examples *****
—
Q: Coal is a fossil fuel that is formed from Answer Choices: (A) water eroding the land. (B) meteors hitting Earth. (C) repeated volcanic eruptions. (D) the decay of organic material.
A:

End your final answer with 'The answer is <answer>.'

Figure 25: 5-shot-restrict prompt for LLM response generation (using alphabet option type question as an example).

You are an expert in commonsense reasoning, Please choose the answer from options A to D, corresponding to the question.

—

***** Start In-Context Examples *****
Q: Which of the following helps to produce urine in humans and other mammals? Answer Choices: (A) bladder (B) urethra (C) kidneys (D) ureter
A: The answer is (C).
Q: An atom consists of a nucleus surrounded by Answer Choices: (A) ions. (B) protons. (C) neutrons. (D) electrons.
A: The answer is (D).
Q: Which lists the diameter of the planets in order from smallest to largest? Answer Choices: (A) Venus, Earth, Mercury, Mars (B) Earth, Mars, Venus, Mercury (C) Mars, Mercury, Earth, Venus (D) Mercury, Mars, Venus, Earth
A: The answer is (D).
Q: In a container, a mixture of water and salt is stirred so that the salt dissolves completely. Sand is added to this solution and allowed to settle to the bottom of the container. If the container is placed on a heat source and the liquid evaporates completely, what will be left in the container? Answer Choices: (A) Nothing will remain in the container. (B) Only salt will remain in the container. (C) Only sand will remain in the container. (D) Salt and sand will both remain in the container.
A: The answer is (D).
Q: Which instrument measures atmospheric pressure? Answer Choices: (A) barometer (B) hygrometer (C) thermometer (D) magnetometer
A: The answer is (A).
***** End In-Context Examples *****

—

Q: Coal is a fossil fuel that is formed from Answer Choices: (A) water eroding the land. (B) meteors hitting Earth. (C) repeated volcanic eruptions. (D) the decay of organic material.
A:

Let's think step by step.

Figure 26: 5-shot-cot prompt for LLM response generation (using alphabet option type question as an example).

You are an expert in commonsense reasoning, Please choose the answer from options A to D, corresponding to the question.

—

***** Start In-Context Examples *****
Q: Which of the following helps to produce urine in humans and other mammals? Answer Choices: (A) bladder (B) urethra (C) kidneys (D) ureter
A: The answer is (C).
Q: An atom consists of a nucleus surrounded by Answer Choices: (A) ions. (B) protons. (C) neutrons. (D) electrons.
A: The answer is (D).
Q: Which lists the diameter of the planets in order from smallest to largest? Answer Choices: (A) Venus, Earth, Mercury, Mars (B) Earth, Mars, Venus, Mercury (C) Mars, Mercury, Earth, Venus (D) Mercury, Mars, Venus, Earth
A: The answer is (D).
Q: In a container, a mixture of water and salt is stirred so that the salt dissolves completely. Sand is added to this solution and allowed to settle to the bottom of the container. If the container is placed on a heat source and the liquid evaporates completely, what will be left in the container? Answer Choices: (A) Nothing will remain in the container. (B) Only salt will remain in the container. (C) Only sand will remain in the container. (D) Salt and sand will both remain in the container.
A: The answer is (D).
Q: Which instrument measures atmospheric pressure? Answer Choices: (A) barometer (B) hygrometer (C) thermometer (D) magnetometer
A: The answer is (A).
***** End In-Context Examples *****

—

Q: Coal is a fossil fuel that is formed from Answer Choices: (A) water eroding the land. (B) meteors hitting Earth. (C) repeated volcanic eruptions. (D) the decay of organic material.
A:

End your final answer with 'The answer is <answer>.'

Let's think step by step.

Figure 27: 5-shot-cot-restrict prompt for LLM response generation (using alphabet option type question as an example).

## F.2 PROMPTS FOR AUTO LABELING USING GPT-4

As shown in Step 3 of Figure 3, we employed a self-consistency strategy for the automatic annotation using GPT-4. Specifically, we used two forms of prompts: a normal form and an XML form, as shown in Figure 28 and Figure 29, respectively. In these figures, the red part represents the System prompt, the orange part represents the Instruction prompt, and the black part represents the Question.

---

You are a help assistant tasked with extracting the precise key answer from given output sentences. You must only provide the extracted key answer without including any additional text.

—

I will provide you with a question, output sentences along with an answer range. The output sentences are the response of the question provided. The answer range could either describe the type of answer expected or list all possible valid answers. Using the information provided, you must accurately and precisely determine and extract the intended key answer from the output sentences. Please don't have your subjective thoughts about the question.

If the output sentences present multiple different answers, carefully determine if the later provided answer is a correction or modification of a previous one. If so, extract this corrected or modified answer as the final response. Conversely, if the output sentences fluctuate between multiple answers without a clear final answer, you should output [No valid answer].

For a question of type "classify_text", if the output sentences do not clearly provide any answer within the specified answer range, then the output should be [No valid answer].

Additionally, if the answer range is a list and the answer given in the output sentences is not included in the answer range, also output [No valid answer].

—

Question: {question}

Output sentences: {LLM response}

Key answer type: {key answer type}

Answer range: {answer range}

Key extracted answer:

Figure 28: The prompt for automatic annotation by GPT-4: normal form.

You are a help assistant tasked with extracting the precise key answer from given output sentences. You must only provide the extracted key answer without including any additional text.
—

I will provide you with a question, output sentences along with an answer range. The output sentences are the responses to the question provided. The answer range could either describe the type of answer expected or list all possible valid answers. Using the information provided, you must accurately and precisely determine and extract the intended key answer from the output sentences. Please don't have your subjective thoughts about the question.
If the output sentences present multiple different answers, carefully determine if the later provided answer is a correction or modification of a previous one. If so, extract this corrected or modified answer as the final response. Conversely, if the output sentences fluctuate between multiple answers without a clear final answer, you should output [No valid answer].
For a question of type "classify_text", if the output sentences do not clearly provide any answer within the specified answer range, then the output should be [No valid answer]. Additionally, if the answer range is a list and the answer given in the output sentences is not included in the answer range, also output [No valid answer].
—

```xml
<?xml version="1.0" ?>
<information>
    <question type="str">{question}</question>
    <output_sentences type="str">{LLM response}</output_sentences>
    <key_answer_type type="str">{key answer type}</key_answer_type>
    <answer_range type="dict">
        <type type="str">a list of all possible valid answers</type>
        <choices type="list">{answer range}</choices>
    </answer_range>
</information>
```

Key extracted answer:

Figure 29: The prompt for automatic annotation by GPT-4: XML form.

## F.3 PROMPTS FOR JUDGE MODELS USING GPT-4

As shown in Figure 30, this is the prompt used by GPT-4 as Judge in our experiments. This prompt is also applied to other judge models used for comparison in our study. Figure 31 illustrates the CoT prompt used by GPT-4 as Judge (CoT).

You are a diligent and precise assistant tasked with evaluating the correctness of responses.

—

We request your feedback on whether the model's response correctly answers the user question above. Based on the reference answer, please determine if the model's answer is correct.
Please first output "correct" or "incorrect" on a single line. In the subsequent line, provide a brief explanation of your judgment, ensuring it is objective and clear.

—

Question: {question}

Reference Answer: {reference}

Model's Answer: {answer}

Figure 30: The prompt for GPT-4 as Judge.

You are a diligent and precise assistant tasked with evaluating the correctness of responses. Think step by step as you make your evaluation.

—

We request your feedback on whether the model's response correctly answers the user question above. Follow these steps to make your evaluation:
1. Think step by step: Read the user question carefully.
2. Think step by step: Review the reference answer and understand the key points it covers.
3. Think step by step: Compare the model's answer with the reference answer.
4. Think step by step: Determine if the model's answer addresses the key points in the reference answer and correctly answers the question.

First, provide your reasoning in detail. Then, clearly state your judgement as either "Correct" or "Incorrect."

Please present your response in the following JSON format:
{{
"reasoning": "Your step-by-step reasoning here.",
"judgement": "Correct or Incorrect"
}}

—

Question: {question}

Reference Answer: {reference}

Model's Answer: {answer}

Figure 31: The prompt for GPT-4 as Judge (CoT).

### F.4 PROMPTS FOR EXTRACTING KEY ANSWERS USING XFINDER

As shown in Figure 32, this is the carefully designed prompt template we used for extracting key answers with xFinder.

You are a help assistant tasked with extracting the precise key answer from given output sentences. You must only provide the extracted key answer without including any additional text.

—

I will provide you with a question, output sentences along with an answer range. The output sentences are the response of the question provided. The answer range could either describe the type of answer expected or list all possible valid answers. Using the information provided, you must accurately and precisely determine and extract the intended key answer from the output sentences. Please don't have your subjective thoughts about the question. First, you need to determine whether the content of the output sentences is relevant to the given question. If the entire output sentences are unrelated to the question (meaning the output sentences are not addressing the question), then output [No valid answer]. Otherwise, ignore the parts of the output sentences that have no relevance to the question and then extract the key answer that matches the answer range. Below are some special cases you need to be aware of:
(1) If the output sentences present multiple different answers, carefully determine if the later provided answer is a correction or modification of a previous one. If so, extract this corrected or modified answer as the final response. Conversely, if the output sentences fluctuate between multiple answers without a clear final answer, you should output [No valid answer].
(2) If the answer range is a list and the key answer in the output sentences is not explicitly listed among the candidate options in the answer range, also output [No valid answer].

—

Question: {question}

Output sentences: {LLM response}

Key answer type: {key answer type}

Answer range: {answer range}

Key extracted answer:

Figure 32: The prompt for extracting key answers by xFinder.

