# OpenReview forum: "xFinder: Large Language Models as Automated Evaluators for Reliable Evaluation"
_ICLR.cc/2025/Conference — ICLR 2025 Poster_

### Official Review · Reviewer_XqZp · 2024-10-29

**Soundness:** 4
**Presentation:** 3
**Contribution:** 4
**Rating:** 6
**Confidence:** 5

**Summary:**

The paper proposes a training dataset KAF for key answer extraction and the correspondingly trained models, xFinder. The motivation lies in improving the extraction of answers in LLM responses for more reliable evaluation. The authors create KAF based on various LLMs, different evaluation tasks, and widely used prompting techniques, i.e., CoT and few-shot prompting. Based on the comprehensive experiments in the paper, xFinder outperforms RegX and other LLM-based methods in answer extraction.

**Strengths:**

The paper is solid enough on all claims with its comprehensive experiments. The proposed KAF dataset is suitable for future research on developing more reliable evaluation systems. The xFinder models are more efficient and reliable than current LLM-based methods. In all, the paper did a good engineering research on using LLMs to better find the answers from their own responses.

**Weaknesses:**

- **Missing Related Work**: The work [1] is also highly related to this work, especially for the Judgement Accuracy part. It holds a similar idea by comparing different evaluation methods, including LLM-based ones, in directly evaluating open-question answering.

- **Annotation Agreement**: Human rechecking is one of the significant parts of the data generation pipeline. What are the annotation agreements between annotators?

- **Human/Case Study**: Except for those numbers in the experiment tables, it is important to do the case study on the output of xFinders. For example, **How and Why is xFinder better**, **Is it worth using xFinder other than RegX or other LLM-based methods?**, **Summarize the failure modes of those inferior methods and how xFinder could perform better in these cases.**, etc.

- **Writing**: I recommend adding more content about experimental settings, such as evaluation metrics, baseline models, etc, to the main text. There are too many staff in the Appendix. Things could be clearly explained in the main text for better reading.

(see the below ``Questions`` section for more)

[1] Wang C, Cheng S, Guo Q, et al. Evaluating open-qa evaluation[J]. Advances in Neural Information Processing Systems, 2024, 36.

**Questions:**

- **Why does the key extraction task need to have ``short text`` and ``alphabet option`` categories, as the former could be transformed into the latter?**

- **Does the order of the xticks in the bump charts affect the comparison between ``alphabet option`` and ``short text``?**

- **What is the trade-off between xFinder and simply adding one or more regular expression patterns in RegX?**: It is known to us that LLMs are good at learning patterns and following instructions, which could have further enhancements as the models' reasoning capabilities are further enhanced. xFinder indeed makes good improvements, but where do they come from? We must be sure that xFinder (or future work) helps answer extraction better than RegX, such as finding answers from those nasty answer patterns. After all, it is easy to write RegX patterns nowadays by simply prompting GPT-4 and LLM-based methods are still inefficient compared to lexical methods.

---

> ### Author Response · Authors · 2024-11-15
> **Response to Reviewer XqZp**
>
> Thank you for your detailed review and constructive feedback on our research work. Below are our detailed responses, which we hope will address your concerns.
>
> ## Answer 1 for Weakness 1
> > Missing Related Work: The work [1] is also highly related to this work ...
>
> Thank you for your valuable suggestion. The paper you recommended is indeed highly valuable, as its proposed EVOUNA dataset makes significant contributions to the automated evaluation of open-ended question-answering tasks. Similar to our work, it aims to enhance the accuracy and reliability of automated evaluations. We have updated the Related Work section of our paper to include **a citation to this work**, further enriching the discussion of related studies.
>
> ## Answer 2 for Weakness 2
> > Annotation Agreement: Human rechecking is ... between annotators?
>
> As you rightly pointed out, manual review is critical for the annotation of the KAF dataset. In **Section 4.2**, we briefly introduced the multi-round annotation and manual review procedures for the KAF dataset. Regarding the inter-annotator agreement, we have provided additional explanations in **Appendix B.2** of the revised paper. Furthermore, the **annotation guidelines** have been included in **Appendix B.2.1** of the updated version.
>
> ## Answer 3 for Weakness 3
> > Human/Case Study: Except for those numbers in the experiment tables ...
>
> Thank you for your suggestion. While our paper provides a detailed performance comparison between xFinder and other methods, conducting further case studies is indeed valuable. In fact, we have presented some **failure cases** of RegEx-based evaluation frameworks in **Figure 2** and **Appendix D.1** of the paper. Additionally, in **Section 3**, we formally defined xFinder's approach to these tasks to facilitate a more systematic understanding of its performance.
>
> For **extraction paradigms** based on LLMs, such as GPT-4, our analysis identified three main **failure modes**:
>
> - Despite being explicitly prompted to extract key answers from the LLM output, GPT-4 often attempts to directly answer the evaluation question itself, leading to extraction errors.
> - Some LLMs exhibit weaker instruction-following capabilities and include multiple key answers in their outputs for evaluation tasks. In such cases, we label the response as "[No valid answer]." However, GPT-4 may still extract one of the key answers, resulting in incorrect extraction.
> - Although GPT-4 has relatively strong **instruction-following abilities**, its extracted content can occasionally contain redundancies, leading to mismatches with the Gold Label. For instance, when the expected Gold Label is "A," GPT-4 might extract "(A) computer savvy."
>
> The Judge Model approach combines both extraction and matching steps. This multi-step reasoning poses challenges for LLMs [2]. Furthermore, Judge Models directly determine the correctness of the evaluated LLM's output or provide a score, which can introduce reliability issues. Even when a Judge Model produces a correct evaluation result, analysis might reveal that it does not genuinely understand the evaluated question, and its result could simply be a random guess, coincidentally correct.
>
> To address these issues, we **decompose the evaluation process** into two distinct steps: key answer extraction and matching. By designing the KAF dataset and xFinder model, we aimed to enhance the reliability of the evaluation process. Experimental results demonstrate the effectiveness of this approach.
>
> ## Answer 4 for Weakness 4
> > Writing: I recommend adding more content about experimental settings ...
>
> Thank you for your suggestion. The **first paragraph of Section 5** in the original paper provides a description of the experimental setup. However, due to the extensive scope of our experiments, we provided additional details in the appendix for reference. To improve readability, we have **revised the first paragraph of Section 5** in the latest version of the paper to more clearly explain the experimental setup (the revised content is highlighted in blue).

---

> ### Author Response · Authors · 2024-11-15
> **(Continued) Response to Reviewer XqZp**
>
> ## Answer 5 for Question 1
> > Why does the key extraction task need to have short text and alphabet option categories ...
>
> We specifically included both short text and alphabet option settings because restricting evaluation tasks solely to the alphabet option may introduce certain reliability issues (refer to **lines 51–68** of the paper). We explored and validated these issues in our study (refer to **lines 523–528** of the paper). Moreover, this dual-setting approach extends the applicability of xFinder and helps enhance the reliability of evaluation results. Additionally, other studies have also investigated and analyzed the potential impact of using the alphabet option on the reliability of evaluation results [3, 4].
>
> ## Answer 6 for Question 2
> > Does the order of the xticks in the bump charts affect the comparison between alphabet option and short text?
>
> Thank you for your question! The **xticks** are solely used to represent the ranking positions of different evaluation methods on the 10 tested LLMs, and **their order does not affect the comparison results between alphabet option and short text**. The task format (e.g., alphabet option and short text) itself influences the rankings, which are independent of the xticks' order.
>
> ## Answer 7 for Question 3
> > What is the trade-off between xFinder and simply adding one or more regular expression patterns in RegEx ...
>
> Regarding the trade-offs between xFinder and RegEx-based methods, it is true that strong LLMs have advantages in learning patterns and executing instructions, but many weaker LLMs still exhibit limited instruction-following capabilities. Moreover, it is important to distinguish between the evaluation performance of LLMs and their instruction-following abilities. The improvements brought by xFinder primarily stem from its adaptability to complex answer patterns and its generalizability, rather than relying solely on specific pattern recognition.
>
> First, while RegEx can perform well with simple and fixed patterns, it often requires **manual creation** of specific patterns for various unique cases when dealing with the **complex and irregular answer formats** of many LLM responses. This approach is not practical and lacks scalability. In contrast, xFinder leverages the semantic understanding capabilities of LLMs to extract key answers from context, demonstrating strong generalization abilities even in the face of highly variable answer patterns. Additionally, xFinder goes beyond simple pattern matching by understanding the relationship between questions and answers, thereby improving Extraction Accuracy. Compared to the pattern-dependent RegEx methods, xFinder demonstrates higher reliability and evaluation accuracy, as evidenced by our experimental results.
>
> In terms of **efficiency**, we recognize that LLM-based methods incur higher computational costs compared to RegEx-based approaches. However, xFinder's advantage lies in reducing the need for manual adjustments and minimizing dependence on **complex regular expressions**. This is especially beneficial in scenarios where answer patterns are diverse, as it effectively improves Extraction Accuracy and enhances the reliability of evaluation results. Furthermore, we analyzed **xFinder’s evaluation efficiency** (refer to **Table 5** in the paper), showing that it can evaluate 200 samples in an average of just 10.67 seconds. While this efficiency still falls short of RegEx-based methods, it surpasses existing Judge Models and is sufficient to meet the needs of most researchers and evaluators.
>
> Thank you again for your valuable feedback. If you have any further questions or comments, please let us know at any time.
>
> [1] Wang, C., Cheng, S., Guo, Q., et al. (2024). Evaluating Open-QA Evaluation. Advances in Neural Information Processing Systems, 36.
>
> [2] Yang, S., et al. (2024). Do Large Language Models Latently Perform Multi-Hop Reasoning? ACL 2024. Retrieved from https://aclanthology.org/2024.acl-long.550.
>
> [3] Balepur, N., Ravichander, A., Rudinger, R. (2024). Artifacts or Abduction: How Do LLMs Answer Multiple-Choice Questions Without the Question? arXiv preprint arXiv:2402.12483.
>
> [4] Li, J., Hu, R., Huang, K., et al. (2024). PertEval: Unveiling Real Knowledge Capacity of LLMs with Knowledge-Invariant Perturbations. arXiv preprint arXiv:2405.19740.

---

> > ### Comment · Reviewer_XqZp · 2024-11-20
> > **Reviewer Response**
> >
> > The author's responses have resolved my concerns. The rating is updated.

---

> > > ### Author Response · Authors · 2024-11-22
> > > **Thanks for the Positive Feedback to Reviewer XqZp**
> > >
> > > Dear Reviewer XqZp,
> > >
> > > Thanks for your positive feedback. We're glad our response addressed your concerns.
> > >
> > > We appreciate your time and insightful suggestions.
> > >
> > > Sincerely,
> > >
> > > The Authors

---

### Official Review · Reviewer_vwGU · 2024-11-04

**Soundness:** 3
**Presentation:** 3
**Contribution:** 3
**Rating:** 6
**Confidence:** 4

**Summary:**

The paper introduces xFinder, a tool designed to enhance the accuracy of evaluating large language models by improving key answer extraction and matching. It identifies flaws in current methods like test set leakage and RegEx limitations, proposes a new dataset (KAF) for training, and shows xFinder outperforms traditional and other automated judge models in extraction and judgment accuracy, thus contributing to more reliable LLM evaluations.

**Strengths:**

- The significance of accurate answer extraction in evaluations is often underestimated, yet it critically impacts results. This study rightly emphasizes this aspect.
- xFinder demonstrates strong performance in accuracy over conventional RegEx frameworks.
- Both the model and its dataset are immediately usable for enhancing the reliability of LLM assessments.
- The paper effectively outlines the problems in current evaluation methods and introduces a well-structured solution.

**Weaknesses:**

- The techniques may not be applicable to responses where the answer is not a short, extractable phrase.
- Although the results are promising, I suspect the technique might be replaced by stronger LLMs used as judges with improved prompting techniques in the near future, which could also generalize better for longer responses. The results in Table 3 are good, showing that even GPT-4 as a judge does not perform as well as xFinder. Therefore, I believe xFinder remains useful at this moment for tasks that have a similar distribution to its training data. It will also be interesting to discuss the combination of xFinder and other techniques.

**Questions:**

What is the prompt you used for GPT-4 as Judge (CoT)?

---

> ### Author Response · Authors · 2024-11-15
> **Response to Reviewer vwGU**
>
> Thank you for acknowledging our work and providing valuable suggestions. Below are detailed responses to your specific comments and questions, aiming to address your concerns.
>
> ## Answer 1 for Weakness 1
> > The techniques may not be applicable to responses where the answer is not a short, extractable phrase.
>
> As you correctly pointed out, the primary motivation behind designing xFinder is to address the unreliability of evaluation results caused by inaccurate answer extraction in LLM evaluations. Therefore, the first version of xFinder was designed to support four major types of mainstream evaluation tasks: alphabet option, short text option, categorical label, and Math option.
>
> At the same time, we recognize that these task types may not fully encompass all real-world application scenarios. However, we believe that xFinder has strong transferability. In future work, we plan to enhance the diversity of the dataset to include a broader range of task types. For example, we aim to incorporate tasks such as long-text comprehension and generation, open-ended questions, and complex reasoning tasks. By adding these new task types, we hope to further improve the applicability and versatility of xFinder.
>
> ## Answer 2 for Weakness 2
> > Although the results are promising, I suspect the technique might be replaced by stronger LLMs used as judges ...
>
> Thank you for your valuable feedback! We understand that future, more advanced LLMs as Judges, combined with improved prompting techniques, may further enhance evaluation performance. However, as shown in **Table 3** of the paper, the accuracy of GPT-4 as a Judge in our experiments is still **significantly lower** than that of xFinder. While LLMs continue to improve, their use as Judges **remains limited by efficiency and cost constraints**. Even with excellent performance, their evaluation efficiency is relatively low, and the high computational cost makes them less practical for evaluators.
>
> In contrast, our **efficiency analysis** demonstrates that xFinder achieves high accuracy while maintaining low evaluation costs. On average, **xFinder evaluates 200 samples in just 10.67 seconds** (refer to **Tables 5 and 6** for efficiency and cost analysis).
>
> Additionally, as you noted, as a contribution to the field of datasets and benchmarks, the KAF dataset and xFinder provide a solid foundation for reliable evaluation and can serve as a valuable resource for future research on automated evaluation methods. We also fully agree on the potential of combining xFinder with other techniques and plan to explore broader application scenarios in the future to further enhance its generality and practicality.
>
> ## Answer 3 for Question 1
> > What is the prompt you used for GPT-4 as Judge (CoT)?
>
> Thank you for your question! We apologize for the oversight in the initial draft, where we omitted the prompt details for GPT-4 as Judge. In the latest version of the paper, we **have added the prompts** used for both GPT-4 as Judge and GPT-4 as Judge (CoT) in **Appendix F.3**.
>
> Thank you again for your valuable feedback. If you have any further questions or comments, please let us know at any time.

---

> > ### Author Response · Authors · 2024-11-22
> > **A Gentle Remind to Reviewer vwGU**
> >
> > Dear Reviewer vwGU,
> >
> > This is a gentle reminder that the response phase is nearing its conclusion, with only a few days remaining. We hope our responses have adequately addressed your questions. If you have any further concerns or would like to discuss anything in the remaining time, we would be more than happy to engage with you.
> >
> > Thank you once again for your valuable time and feedback.
> >
> > Sincerely,
> >
> > The Authors

---

> > > ### Comment · Reviewer_vwGU · 2024-11-22
> > >
> > > Thank you for your response. The current prompt for GPT-4 as a Judge (CoT) seems suboptimal. The used prompt is:
> > >
> > > ...
> > > Please first output "Correct" or "Incorrect" on a single line. In the subsequent line, provide a brief
> > > explanation of your judgment, ensuring it is objective and clear
> > > ...
> > >
> > > A better design would let the model generate its rationale first, followed by outputting "Correct" or "Incorrect." Additionally, you might consider having the model generate its output in a structured format like JSON or XML for easier parsing.
> > >
> > > Apologies for the delayed reply; I have been very busy lately. However, I am very interested in seeing GPT-4's performance with the suggested prompt if the authors have time.

---

> > > > ### Author Response · Authors · 2024-11-23
> > > > **Response on GPT-4 as a Judge (CoT) Prompt Design (Reviewer vwGU)**
> > > >
> > > > Thank you for the follow-up and your thoughtful suggestions regarding the prompt design for GPT-4 as a Judge (CoT).
> > > >
> > > > Based on your suggestions, we optimized the prompt template for GPT-4 as Judge (CoT) and conducted additional experiments. Furthermore, we included experimental results for GPT-4o as Judge (CoT).
> > > >
> > > > The results show that, with the CoT-V2 prompt, the Judgement Accuracy of GPT-4 as Judge (CoT) reached 88.42%, an improvement of 5.27% compared to the initial CoT prompt. Meanwhile, the Judgement Accuracy of GPT-4o as Judge (CoT) was 93.2%, though it is still 4.41% lower than xFinder-llama38it. These findings validate your point that more precise and detailed prompt templates, as well as more advanced LLMs, can indeed enhance performance.
> > > >
> > > > However, it is important to note that using these advanced LLMs as judge models or employing more sophisticated prompt templates significantly increases computational costs, as evidenced by the total tokens (i.e., the number of tokens processed for input and output). For example, in our new experiments, GPT-4 with the updated CoT-V2 prompt incurred a total cost of \\$46.82, which is a 65.73\% increase compared to the \\$28.25 cost without CoT, even 38.6\% increase compared to the \\$33.79 cost with CoT-V1. In contrast, although OpenAI has significantly reduced the pricing for GPT-4o, completing the same tasks still costs \\$13.97. It is worth mentioning that this experiment was conducted on a relatively small-scale QA test with only 4k samples, without involving long-context scenarios. However, in real-world evaluation settings, testing often requires significantly larger-scale datasets. This makes these powerful LLMs as judge models impractical for real-world applications.
> > > >
> > > > In summary, for these tasks, xFinder not only achieves higher Judgement Accuracy but also operates at a cost far lower than that of current or even foreseeable future strong LLMs with comparable accuracy. This highlights xFinder's superior practicality in resource-constrained real-world applications.
> > > >
> > > > We hope that we addressed your concerns and questions, and we are happy to continue the discussion.
> > > >
> > > > | Method                     | alphabet option | short text | categorical label | math   | Overall | Total tokens | Total costs (in USD) |
> > > > |-|-|-|-|-|-|-|-|
> > > > | GPT-4 as Judge            | 0.9016          | 0.8909     | 0.7294             | 0.9313 | 0.8420  | 2030563      | 28.25               |
> > > > | GPT-4 as Judge (CoT-V1)      | 0.9016          | 0.8846     | 0.7038             | 0.9313 | 0.8315  | 2440424      | 33.79               |
> > > > | GPT-4 as Judge (CoT-V2)   | 0.9234          | 0.9345     | 0.7919             | 0.9609 | 0.8842  | 2953525      | 46.82               |
> > > > | GPT-4o as Judge (CoT-V2)  | 0.9656          | 0.9709     | 0.8662             | 0.9703 | 0.9320  | 2949108      | 13.97               |
> > > > | xFinder-qwen1505          | 0.9781          | 0.9761     | 0.9625             | 0.9969 | 0.9748  | -            | -                   |
> > > > | xFinder-llama38it         | 0.9750          | 0.9688     | 0.9731             | 0.9969 | 0.9761  | -            | -                   |
> > > >
> > > >
> > > > ----
> > > > **CoT-V2 Prompt**
> > > > ```
> > > > You are a diligent and precise assistant tasked with evaluating the correctness of responses. Think step by step as you make your evaluation.
> > > > —
> > > > We request your feedback on whether the model's response correctly answers the user question above. Follow these steps to make your evaluation:
> > > > 1. Think step by step: Read the user question carefully.
> > > > 2. Think step by step: Review the reference answer and understand the key points it covers.
> > > > 3. Think step by step: Compare the model's answer with the reference answer.
> > > > 4. Think step by step: Determine if the model's answer addresses the key points in the reference answer and correctly answers the question.
> > > >
> > > > First, provide your reasoning in detail. Then, clearly state your judgement as either "Correct" or "Incorrect."
> > > >
> > > > Please present your response in the following JSON format:
> > > > {{
> > > >     "reasoning": "Your step-by-step reasoning here.",
> > > >     "judgement": "Correct or Incorrect"
> > > > }}
> > > >
> > > >
> > > > —
> > > > Question: {question}
> > > >
> > > > Reference Answer: {reference}
> > > >
> > > > Model's Answer: {answer}
> > > > ```

---

> > > > > ### Comment · Reviewer_vwGU · 2024-11-24
> > > > >
> > > > > Thank you for the update. Overall, I think my evaluation is fair, and so I keep my rating.

---

> > > > > > ### Author Response · Authors · 2024-11-25
> > > > > >
> > > > > > Dear Reviewer vwGU,
> > > > > >
> > > > > > Thank you for your feedback and taking the time to review our responses! We truly appreciate your comments, time, and patience.
> > > > > >
> > > > > > Sincerely,
> > > > > >
> > > > > > The Authors

---

### Official Review · Reviewer_rbE3 · 2024-11-04

**Soundness:** 3
**Presentation:** 3
**Contribution:** 2
**Rating:** 6
**Confidence:** 4

**Summary:**

This paper introduces xFinder, a novel evaluator designed for answer extraction and matching in the context of LLM evaluation. The study identifies the limitations of current answer extraction modules, particularly those based on RegEx, in handling inconsistencies in model response formats. To address these issues, the authors propose xFinder to enhance extraction accuracy and evaluation reliability. The authors developed a dataset called the Key Answer Finder (KAF) dataset to train xFinder. Experimental results demonstrate that xFinder significantly outperforms existing frameworks and model-based evaluators in terms of extraction accuracy and evaluation efficiency.

**Strengths:**

The identified challenges in LLM response extraction and matching are realistic and merit attention.

The paper proposes a novel method to improve answer extraction modules, addressing limitations in existing approaches.

The paper is well-structured, with a clear progression from problem definition to methodology and experimental analysis.

**Weaknesses:**

Although the KAF dataset is used to validate xFinder’s performance, the paper lacks a comprehensive exploration of the model’s generalizability to entirely different datasets and error types.

The paper does not include sufficient experimental analysis on the impact of xFinder on final evaluation outcomes, such as a comparison between using xFinder and other extraction methods in terms of evaluation results.

There is a lack of detailed analysis of the KAF dataset’s quality, such as inter-annotator agreement metrics.

**Questions:**

see weaknesses

---

> ### Author Response · Authors · 2024-11-15
> **Response to Reviewer rbE3**
>
> Thank you for your detailed review and valuable comments. Below are detailed responses to your concerns.
>
> ## Answer 1 for Weakness 1
> > Although the KAF dataset is used to validate xFinder’s performance, the paper lacks... generalizability to... different datasets and error types.
>
> We would like to clarify our exploration of xFinder's generalization ability across **different datasets**. In the KAF dataset, we **specifically designated a Generalization Set** to evaluate xFinder's generalization capabilities. For this Generalization Set, we used evaluation datasets completely different from the Training Set used for fine-tuning xFinder (e.g., OpenbookQA and SIQA). Additionally, we incorporated responses generated by various LLMs (e.g., Llama3-8B-Instruct and Qwen1.5-MoE-A2.7B-Chat) and designed multiple distinct prompting templates. These efforts ensured that the Generalization Set contained a diverse range of LLM responses.
>
> This setup provided a comprehensive test of xFinder's ability to generalize across different datasets and various LLM responses. As shown in **Table 2** of the paper, xFinder's performance on the Generalization Set demonstrates its **strong generalization ability**. Details about the composition of each part of the dataset are provided in **Appendix B.1**.
>
> ## Answer 2 for Weakness 2
> > The paper does not include sufficient analysis on the impact of xFinder on... outcomes, such as a comparison with other extraction methods.
>
> Regarding xFinder's impact on final evaluation results, we conducted **extensive experiments** in our paper. In **Table 3**, we compared the Judgement Accuracy of xFinder, other extraction-based LLM evaluation frameworks, Judge Models, and GPT-4. The results show that among the extraction-based frameworks, the highest Judgement Accuracy achieved by OpenCompass is **only 88.7%**, while PandaLM's Judgement Accuracy is **51.9%**. The 33B JudgeLM achieves a Judgement Accuracy of **78.13%**, and GPT-4 as a Judge achieves **84.2%**, both significantly **lower than the 97.61% achieved by xFinder**.
>
> In **Table 4**, we analyzed the **discrepancies between Extraction Accuracy and Judgement Accuracy** across various methods. The baseline method with the smallest discrepancy, OpenCompass, has a **14.32% gap** between its Judgement Accuracy and Extraction Accuracy, highlighting the **significant unreliability** of traditional RegEx-based extraction methods. In contrast, our xFinder-llama38it shows a gap of **only 2.43%** between its Judgement Accuracy and Extraction Accuracy, effectively reducing errors caused by inaccurate answer extraction.
>
> Additionally, in **Section 5.3**, we presented the evaluation results of different extraction-based frameworks across various datasets (further comparisons between xFinder and baseline methods are provided in **Appendix D.3**). The results demonstrate **significant inconsistencies** among the evaluation outcomes produced by different frameworks, indicating the unreliability of current LLM evaluation results and corresponding leaderboards. These findings further highlight the robustness and reliability of xFinder's evaluation results.
>
> ## Answer 3 for Weakness 3
> > There is a lack of detailed analysis of the KAF dataset’s quality, such as inter-annotator agreement metrics.
>
> To ensure the quality of the KAF dataset, we implemented rigorous multi-round annotation and manual review procedures, employing different annotation strategies for the Training Set, Test Set, and Generalization Set.
>
> 1. **Training Set:** To enhance annotation efficiency, we adopted a semi-automated annotation strategy. Specifically, we used GPT-4 with different prompts (refer to Appendix F.2) to generate two sets of annotations. We then applied the Self-Consistency approach to identify data items with inconsistent annotations. These items were subsequently manually annotated to ensure accuracy.
>
> 2. **Test Set and Generalization Set:** For these sets, we conducted two rounds of manual annotation on all data items to ensure label accuracy and consistency.
>
> For data that requires manual annotation, each data item is annotated by two different annotators. If the two annotators produce different results for the same item, the authors will recheck the annotations and make the final decision.
>
> Details about the **dataset annotation procedures** can be found in Section 4.2 of the paper. Additionally, we have included **guidelines on manual annotation** in Appendix B.2.1 of the revised version of the paper. We believe that these measures effectively ensure the quality of the KAF dataset, providing a reliable foundation for training and evaluating xFinder.
>
> Thank you again for your valuable feedback. If you have any further questions or comments, please let us know at any time.

---

> > ### Author Response · Authors · 2024-11-22
> > **A Gentle Remind to Reviewer rbE3**
> >
> > Dear Reviewer rbE3,
> >
> > This is a gentle reminder that the response phase is nearing its conclusion, with only a few days remaining. We hope our responses have adequately addressed your questions. If you have any further concerns or would like to discuss anything in the remaining time, we would be more than happy to engage with you.
> >
> > Thank you once again for your valuable time and feedback.
> >
> > Sincerely,
> >
> > The Authors

---

> > > ### Author Response · Authors · 2024-11-25
> > > **A Gentle Follow-up Reminder to Reviewer rbE3**
> > >
> > > Dear Reviewer rbE3,
> > >
> > > This is a gentle reminder that the response phase is nearing its conclusion, with only about two days remaining. We hope we have adequately addressed your questions. If your concerns have been resolved, we kindly ask if you could consider revisiting your review score. If not, please feel free to let us know, as we are more than willing to provide further clarification.
> > >
> > > Thank you again for your valuable time and thoughtful review.
> > >
> > > Sincerely,
> > >
> > > The Authors

---

> > ### Comment · Reviewer_rbE3 · 2024-11-25
> > **Reply by Reviewer rbE3**
> >
> > Thanks for your detailed response. I have read and changed my score accordingly.

---

> > > ### Author Response · Authors · 2024-11-26
> > >
> > > Dear Reviewer rbE3,
> > >
> > > Thanks for your positive feedback. We're glad our response addressed your concerns.
> > >
> > > We appreciate your time and insightful suggestions.
> > >
> > > Sincerely,
> > >
> > > The Authors

---

### Official Review · Reviewer_B8RQ · 2024-11-04

**Soundness:** 3
**Presentation:** 3
**Contribution:** 3
**Rating:** 6
**Confidence:** 3

**Summary:**

This paper proposes a novel evaluator for answer extraction and matching in LLM evaluation. The main idea is to first construct a large-scale LLM response evaluation dataset, and then train (small) LLMs on it. This paper conducts an extensive evaluation of multiple tasks with comparison with multiple LLM-based evaluators.

**Strengths:**

1 A large dataset that can be used for further LLM-based evaluation.
2 A new model that can be used for more reliable evaluation.

**Weaknesses:**

To be perfectly honest, I am not an expert in LLM-based evaluation. But to me, the main contribution is the construction of a dataset that can help train LLM evaluators, with the help of other LLMs (e.g GPT-4). Thus the novelty of the proposed model is less convincing as it does not provide any new architecture. Training LLMs on evaluation data as evaluators have also been explored in previous research, such as [1] and its subsequent work. Could the authors explain more on its novelty? For example, in terms of training process, and model architectures, how does the xFinder differ from previous work that trains LLMs on evaluation data? Also, would it possible to prompt GPT-4 or other very big LLM using ICL with the constructed data, and how would it perform?

[1] Towards a Unified Multi-Dimensional Evaluator for Text Generation

**Questions:**

Please see above.

---

> ### Author Response · Authors · 2024-11-15
> **Response to Reviewer B8RQ**
>
> Thank you for acknowledging our work and providing valuable suggestions. Below are detailed responses to your specific comments and questions, aiming to address your concerns.
>
> ## Answer for Questions
> > To be perfectly honest ... would it possible to prompt GPT-4 or other very big LLM using ICL with the constructed data, and how would it perform?
>
> Thank you for your thoughtful review of our work! We are glad to provide clarifications regarding your concerns about the novelty and comparisons to other LLM evaluation methods.
>
> First and foremost, our study focuses on enhancing the reliability of LLM evaluations rather than proposing a new model architecture. This work falls within **the domain of datasets and benchmarks**, aiming to support the iterative improvement of LLMs through more robust evaluation methods. Through our analysis, we identified **significant reliability issues** in current evaluation frameworks, which can prevent researchers from accurately understanding the limitations of the evaluated LLMs. This, in turn, may lead to **biased advancements in LLM development**. Thus, improving the reliability of LLM evaluations is of paramount importance.
>
> There have been numerous studies proposing **automated evaluation methods** based on Judge Models [1, 2, 3]. However, previous research has highlighted issues with the **generalization ability and fairness** of these fine-tuned Judge Models [4]. Specifically, Judge Models often suffer from **overfitting** to the datasets used for fine-tuning, resulting in biases when applied to real-world evaluation scenarios.
>
> Additionally, the Judge Model approach combines key answer extraction and matching into a single process, introducing a "skipping step" problem. Such multi-step reasoning poses challenges for LLMs [5]. The Judge Model directly determines the correctness or assigns a score, which might be **unreliable**. Even if the Judge Model's evaluation result is correct, we might find that it did not truly understand the problem itself, and the evaluation result might just be a random guess that happened to be correct. The experimental results in **Table 4** of our paper corroborate this issue. For example, when using GPT-4 as a Judge, there is a **significant gap** between its Extraction Accuracy when splitting extraction and matching and its overall Judgement Accuracy when combining these steps.
>
> To address this, we decompose the evaluation process into two steps: key answer extraction and matching. We designed and constructed the KAF dataset and the xFinder model to support this approach. Experimental results demonstrate that xFinder excels in performance, achieving significantly higher Judgement Accuracy than existing Judge Models (refer to **Table 3**). Moreover, **xFinder demonstrates higher evaluation efficiency** compared to current Judge Models (refer to **Table 5**), substantially improving the reliability of LLM evaluations.
>
> Regarding the question of using datasets to prompt models like GPT-4 for evaluation, we conducted relevant comparison experiments, as shown in **Table 3** of our paper. The results indicate that even GPT-4's performance lags significantly behind xFinder (a model with **only 0.5B parameters**). Furthermore, prompting such powerful LLMs as evaluators is **inefficient**, incurs high costs, and is impractical for widespread adoption in real-world evaluation tasks. We believe this further demonstrates the practicality and value of xFinder.
>
> Thank you again for your valuable feedback. If you have any further questions or comments, please let us know at any time.
>
> [1] Zheng, L., et al. (2023). Judging LLM-as-a-Judge with MT-Bench and Chatbot Arena. NeurIPS 2023. Retrieved from https://openreview.net/forum?id=uccHPGDlao.
>
> [2] Wang, Y., et al. (2024). PandaLM: An Automatic Evaluation Benchmark for LLM Instruction Tuning Optimization. ICLR 2024. Retrieved from https://openreview.net/forum?id=5Nn2BLV7SB.
>
> [3] Zhu, L., Wang, X., Wang, X. (2023). JudgeLM: Fine-Tuned Large Language Models are Scalable Judges. arXiv preprint arXiv:2310.17631.
>
> [4] Wang, Y., et al. (2024). Large Language Models are not Fair Evaluators. ACL 2024. Retrieved from https://aclanthology.org/2024.acl-long.511.
>
> [5] Yang, S., et al. (2024). Do Large Language Models Latently Perform Multi-Hop Reasoning? ACL 2024. Retrieved from https://aclanthology.org/2024.acl-long.550.

---

> > ### Comment · Reviewer_B8RQ · 2024-11-21
> >
> > Thank the authors for clarification.
> > Overall, I think my evaluation is fair, and I will keep my score.

---

> > > ### Author Response · Authors · 2024-11-22
> > > **Thanks for the Positive Feedback to Reviewer B8RQ**
> > >
> > > Dear Reviewer B8RQ,
> > >
> > > Thank you for recognizing our responses. We’re pleased to receive your positive feedback! We truly appreciate your comments, time, and patience.
> > >
> > > Sincerely,
> > >
> > > The Authors

---

### Author Response · Authors · 2024-11-20
**A Friendly Reminder to the Reviewers**

Dear Reviewers,

Thank you once again for your thoughtful comments and valuable feedback. To address your comments and suggestions, we have submitted our responses and revised manuscript, with the revised sections highlighted in blue font. Since the discussion phase is halfway through, we kindly request the reviewers to confirm receipt of our responses. We also welcome any further concerns or suggestions regarding our responses.

Thank you for your time and consideration.

Sincerely,

The Authors

---

### Author Response · Authors · 2024-11-29
**Global Response to Area Chairs and Reviewers**

Dear Area Chairs and Reviewers,

We deeply appreciate your thoughtful feedback and the time you’ve invested in reviewing our work. Your insights during both the review and discussion phases have been invaluable in enhancing the quality of our paper.

In response to the reviewers' concerns and suggestions, we have provided detailed explanations or clarifications in the discussion and submitted a revised manuscript. All updates are highlighted in blue font for easy reference, and the main revisions include the following four improvements:

1. **Additional analysis of related work** [Reviewer XqZp] (see Lines 163-165 of the manuscript).
2. **More detailed manual annotation protocol** [Reviewer XqZp, Reviewer rbE3] (see Appendix B.2).
3. **Improved descriptions of the experimental setup** [Reviewer XqZp] (see Lines 326-347 of the manuscript).
4. **Prompt for GPT-4 as Judge (CoT)** [Reviewer vwGU] (see Appendix F.3).

Additionally, we noticed that **the ICLR discussion period has been extended to December 3, 2024 (AoE).** Given this change in the review process and its potential impact on the evaluation standards, we would like to take this opportunity to further summarize the core contributions of our paper and reiterate its importance and potential value in the field of reliable LLM evaluation. Our research not only makes significant improvements to existing methods but also provides a unique perspective for the LLM evaluation domain:

- **Significance of the Work:** We analyzed the existing LLM evaluation pipeline and introduced the concept of **reliable evaluation**. By examining critical factors that compromise evaluation reliability, particularly in answer extraction and matching stages, we identified challenges that hinder accurate model evaluation and model improvement. Enhancing evaluation reliability is essential for advancing LLM research.

- **Contributions and Impact:** We constructed the high-quality KAF dataset for training and evaluating LLMs as automated evaluators, and developed xFinder to automate answer extraction and matching. Experiments show that xFinder outperforms existing methods like RegEx and LLM as a judge in judgement accuracy (e.g., xFinder-qwen1505 achieves 97.48% accuracy) and demonstrates stronger generalization across datasets. Additionally, xFinder offers superior efficiency and cost benefits (e.g., the smallest model with only 0.5B parameters), providing an effective, low-cost solution for automated evaluation.

- **Challenges and Our Solution:** Automated evaluation, exemplified by LLM as a judge, has become crucial in LLM evaluation [1, 2, 3]. However, current models combine multiple steps (e.g., answer extraction and matching) in a single process, leading to intermediate skipping issues and lower accuracy [4]. Moreover, these models struggle with generalization, fairness, and efficiency [5]. By separating the evaluation into two stages—answer extraction and matching—xFinder improves judgement accuracy, generalization, and efficiency, thus enhancing the overall reliability and practicality of automated evaluation.

- **Future Applications and Potential:** Beyond automated evaluation, the KAF dataset and xFinder have broader applications in other tasks and domains. For instance, the strategy of separating answer extraction and matching can be applied to automate evaluation in other open-ended question tasks. Additionally, the high-quality annotated KAF dataset can be used in structured text generation tasks. Our analysis of the unreliabilities in LLM evaluation also provides valuable insights for future research and the development of more reliable evaluation methods, contributing to more robust and scalable LLM evaluation frameworks.

Once again, we sincerely thank all reviewers for their detailed and thoughtful feedback on this paper. Most of the suggestions aim to clarify issues or explore further improvements, which we have carefully addressed in individual responses. Additionally, we hope that, with the extension of the discussion period, we can further engage with the reviewers to explore potential issues or optimization directions, and contribute to the ICLR community with a higher-quality paper. If you have any additional suggestions or require further clarification, please do not hesitate to let us know.

Sincerely,

The Authors

---
**References**

[1] Zheng, L., et al. (2023). Judging LLM-as-a-Judge with MT-Bench and Chatbot Arena. *NeurIPS 2023.*

[2] Wang, Y., et al. (2024). PandaLM: An Automatic Evaluation Benchmark for LLM Instruction Tuning Optimization. *ICLR 2024.*

[3] Gu, J., et al. (2024). A Survey on LLM-as-a-Judge. *arXiv preprint arXiv:2411.15594.*

[4] Yang, S., et al. (2024). Do Large Language Models Latently Perform Multi-Hop Reasoning? *ACL 2024.*

[5] Wang, Y., et al. (2024). Large Language Models are not Fair Evaluators. *ACL 2024.*

---

### Meta-Review · Area_Chair_4g7s · 2024-12-23

**Metareview:**

The paper proposed to improve the robustness of answer extraction when evaluating large language models, focusing on the limitations of existing RegEx-based evaluation frameworks. It argues that these methods often lead to extraction errors and unreliable evaluations. The paper proposes xFinder, a novel evaluator designed to enhance the accuracy and reliability of answer extraction and matching. The authors also introduce the KAF dataset to support training and evaluation of the extraction task. Experiments show that xFinder has a significantly higher extraction accuracy compared to RegEx-based methods. It improves final judgment accuracy to 97.61%, outperforming GPT-4.

Strengths

The work studied an important issue in the evaluation of LLMs.
The experimental results demonstrate significant improvements in accuracy and efficiency over existing methods, particularly RegEx and LLMs like GPT-4.
The KAF dataset is a strong contribution, providing a benchmark for future research in automated evaluation.

Weaknesses
While the KAF dataset is valuable, more detailed exploration of its quality (e.g., inter-annotator agreement, error types) could strengthen the paper.
The method may struggle with long or complex responses, as noted by reviewers. This limitation is not explored in depth.
Some details, particularly experimental setups and case studies should be moved out of appendix.

Given its strengths in addressing an important problem, with its robust empirical results, I recommend acceptant of this paper.

**Additional Comments On Reviewer Discussion:**

Reviewer mentioned the model's novelty is limited since it does not propose a new architecture. The authors clarified that the paper’s primary goal is improving evaluation reliability through structured processes (key answer extraction and matching). This concern was satisfactorily addressed. The explanation reinforced the importance of improving reliability over introducing new architectures.

---

### Decision · Program_Chairs · 2025-01-22

Accept (Poster)